# Oncolytic H-1 parvovirus binds to sialic acid on laminins for cell attachment and entry

Amit Kulkarni [1,2], Tiago Ferreira [1], Clemens Bretscher[1], Annabel Grewenig[1], Nazim El-Andaloussi[1,10], Serena Bonifati[1,11], Tiina Marttila[1,2], Valérie Palissot[2], Jubayer A. Hossain[2,3,4], Francisco Azuaje [5,12], Hrvoje Miletic[3,4], Lars A. R. Ystaas[3], Anna Golebiewska [6], Simone P. Niclou [6], Ralf Roeth[7,8], Beate Niesler [7,8], Amélie Weiss[9], Laurent Brino[9] & Antonio Marchini [1,2✉]

H-1 parvovirus (H-1PV) is a promising anticancer therapy. However, in-depth understanding of its life cycle, including the host cell factors needed for infectivity and oncolysis, is lacking. This understanding may guide the rational design of combination strategies, aid development of more effective viruses, and help identify biomarkers of susceptibility to H-1PV treatment. To identify the host cell factors involved, we carry out siRNA library screening using a druggable genome library. We identify one crucial modulator of H-1PV infection: laminin γ1 (*LAMC1*). Using loss- and gain-of-function studies, competition experiments, and ELISA, we validate *LAMC1* and laminin family members as being essential to H-1PV cell attachment and entry. H-1PV binding to laminins is dependent on their sialic acid moieties and is inhibited by heparin. We show that laminins are differentially expressed in various tumour entities, including glioblastoma. We confirm the expression pattern of laminin γ1 in glioblastoma biopsies by immunohistochemistry. We also provide evidence of a direct correlation between *LAMC1* expression levels and H-1PV oncolytic activity in 59 cancer cell lines and in 3D organotypic spheroid cultures with different sensitivities to H-1PV infection. These results support the idea that tumours with elevated levels of γ1 containing laminins are more susceptible to H-1PV-based therapies.

[1] Laboratory of Oncolytic Virus Immuno-Therapeutics, German Cancer Research Center, Heidelberg, Germany. [2] Laboratory of Oncolytic Virus Immuno-Therapeutics, Luxembourg Institute of Health, Luxembourg, Luxembourg. [3] Department of Biomedicine, University of Bergen, Bergen, Norway. [4] Department of Pathology, Haukeland University Hospital, Bergen, Norway. [5] Quantitative Biology Unit, Luxembourg Institute of Health, Luxembourg, Luxembourg. [6] NorLux Neuro-Oncology Laboratory, Department of Oncology, Luxembourg Institute of Health, Luxembourg, Luxembourg. [7] nCounter Core Facility, Institute of Human Genetics, University of Heidelberg, Heidelberg, Germany. [8] Department of Human Molecular Genetics, University of Heidelberg, Heidelberg, Germany. [9] Institut de Génétique et de Biologie Moléculaire et Cellulaire (IGBMC), Illkirch, France. [10] Lonza Cologne GmbH, Köln, Germany. [11] Center for Retrovirus Research, Department of Veterinary Biosciences, The Ohio State University, Columbus, OH, USA. [12] Genomics England, London, United Kingdom. ✉email: antonio.marchini@lih.lu

Oncolytic viruses (OVs) selectively replicate in and destroy tumour cells without harming normal healthy tissues. They act in a multimodal fashion by inducing lysis of the cells and anticancer immunity[1–4]. More than 40 OVs from at least nine virus families are currently being tested against various malignancies in early- or late-phase clinical trials. In addition, the engineered herpes simplex virus encoding granulocyte–macrophage colony-stimulating factor (talimogene laherparepvec, Imlygic™) was granted approval in 2015 both in the USA and in Europe for use against malignant metastatic melanoma[5]. There is optimism that other OVs may be approved in the near future for the treatment of other cancers[6]. However, OVs as a standalone therapy have rarely been reported to induce the complete regression of tumours. Major efforts to improve the clinical outcome of OV treatments are directed towards the search for anticancer modalities that synergise with OVs to kill cancer cells without toxic side-effects. One promising avenue is the combination of OVs with other forms of immunotherapy (e.g., checkpoint blockade)[3,7–11]. Another is to identify patients with tumours whose genetic characteristics are favourable to the virus life cycle and who are thus most likely to benefit from OV treatment. This could lead to the design of 'smart' clinical trials that reduce clinical costs and approval times. A better understanding of the OV life cycle and the identification of host cellular determinants that contribute to virus infection are crucial to guide both the rational design of combination treatments and the identification of biomarkers that could be used for patient stratification.

Rat protoparvovirus H-1PV is a clinically relevant OV whose anticancer potential has been demonstrated at the preclinical level in a number of in vitro cell systems and animal models[12,13]. The first phase I/IIa trial in patients with recurrent glioblastoma (GBM) showed that H-1PV treatment as a standalone therapy is safe, well tolerated and associated with first evidence of efficacy, including: (i) ability to cross the blood–brain (tumour) barrier after intravenous delivery; (ii) widespread intratumoural distribution and expression; (iii) immunoconversion of tumour microenvironment; and (iv) extended median progression-free/overall survival in comparison with historical controls[14,15]. A second clinical trial in patients with pancreatic carcinomas is now in its evaluation phase[16].

H-1PV is a small, non-enveloped, single-stranded DNA virus. Its 5.1 kb genome is organised into the NS and VP gene units, whose expression is regulated by the P4 and P38 promoters, respectively. The *NS* gene unit encodes the NS1, NS2 and NS3 proteins, whereas the *VP* gene unit encodes the VP1 and VP2 capsid proteins and the SAT non-structural protein. NS1 is a multifunctional protein that regulates virus DNA replication and gene transcription, and it is the major effector of H-1PV oncolysis[13,17,18].

H-1PV's DNA replication and gene expression depend on host cell factors. For instance, replication relies on the E2 family of transcription factors, cAMP response element binding protein, activating transcription factors and cyclin A[12,17], which are normally overexpressed in fast-proliferating cancer cells and are therefore important determinants of virus oncotropism. In addition, NS1 activity is modulated by post-translational modifications such as phosphorylation and acetylation[17,19,20].

Many of the host cell factors that have a role in the H-1PV life cycle are yet to be identified. It is largely unknown why some cancer cell lines are highly susceptible to H-1PV infection, whereas others derived from the same tumour entity are less sensitive or even completely refractory. Differences in permissiveness are thought to be regulated by cell–host interactions. Non-permissive tumour cells may lack important factors needed

for different stages of the virus life cycle, e.g., virus cell attachment and entry.

The extracellular matrix (ECM) is a three-dimensional interlocking mesh of extracellular macromolecules, encompassing laminins, fibronectin, collagen, elastin, heparan sulphate, chondroitin sulphate, keratan sulphate and hyaluronic acid, which provide structural and biochemical support to the surrounding cells. The ECM often represents a barrier to a virus on its way into the host cell. Some viruses (and other microorganisms) have developed strategies to bind to specific ECM components[21,22]. This leads to the accumulation of a large quantity of virus particles at the host cell surface, thereby facilitating efficient engagement with host cell transmembrane receptor(s) or receptor complex(es). The attachment to ECM proteins is therefore an essential event that primes the virus to recognise cell surface molecules, initiates virus infection, and hence represents a key determinant of virus tropism and infectivity. Although attachment factors and functional receptors have been identified for some of the members of the Parvoviridae family (e.g., transferrin receptor for the canine and feline parvoviruses; several receptors and attachment factors for a number of adeno-associated virus serotypes[23–26]), the attachment receptor (complex) involved in H-1PV ECM and cell membrane recognition is still unknown. An essential component needed for H-1PV cell attachment is sialic acid (SA); two residues on the viral capsid are involved in SA interaction[27,28]. This property is shared with other protoparvoviruses, such as the minute virus of mice[29] and porcine parvovirus[30]. However, it remains to be characterised whether SA alone is sufficient to mediate H-1PV cell membrane recognition and entry or if the virus requires additional interaction(s) with other proteins present at the cell surface (sialylated or not). Recently, we showed that after binding to the cell membrane, H-1PV enters cells via clathrin-mediated endocytosis[31], a property shared with other protoparvoviruses[32].

In this study, we searched for novel host cell factors involved in the H-1PV life cycle with a primary focus on the sialylated proteins required for H-1PV attachment at the cellular membrane. We identified laminins as key mediators of virus cell attachment and entry.

## Results

**Identification of putative modulators of the H-1PV life cycle by siRNA library screening.** High-throughput RNA interference (siRNA or small hairpin RNA) library screening is a powerful technology for identifying cell factors that negatively or positively modulate the infectivity of a certain virus in host cells[33]. To identify host factors involved in the oncolytic H-1PV life cycle, we performed a siRNA library screening of the druggable genome in a plate format using the cervical carcinoma-derived HeLa cell line (Fig. 1, see Materials and Methods for details). Two sets of cells were reverse transfected with control siRNAs or the druggable siRNA library. One set of cells was left untreated to control the intrinsic cytotoxicity of every transfected siRNA pool, while the other set of cells was infected with recH-1PV-EGFP to analyse H-1PV transduction. At 24 h after infection, cells were fixed and EGFP signal and cell viability were measured (Fig. 1a, b). Based on the EGFP transduction values, the genes were then classified into three groups: (i) putative H-1PV activator genes, whose silencing decreases the EGFP signal, with 151 genes considered as top activators because their silencing reduced EGFP signalling intensity by >70% in comparison with control siRNA; (ii) putative H-1PV repressor genes, whose silencing enhances the EGFP signal, with 89 genes considered as top repressors because their silencing increased EGFP signalling intensity by >160%; and (iii) unrelated genes, whose silencing did not affect EGFP signal

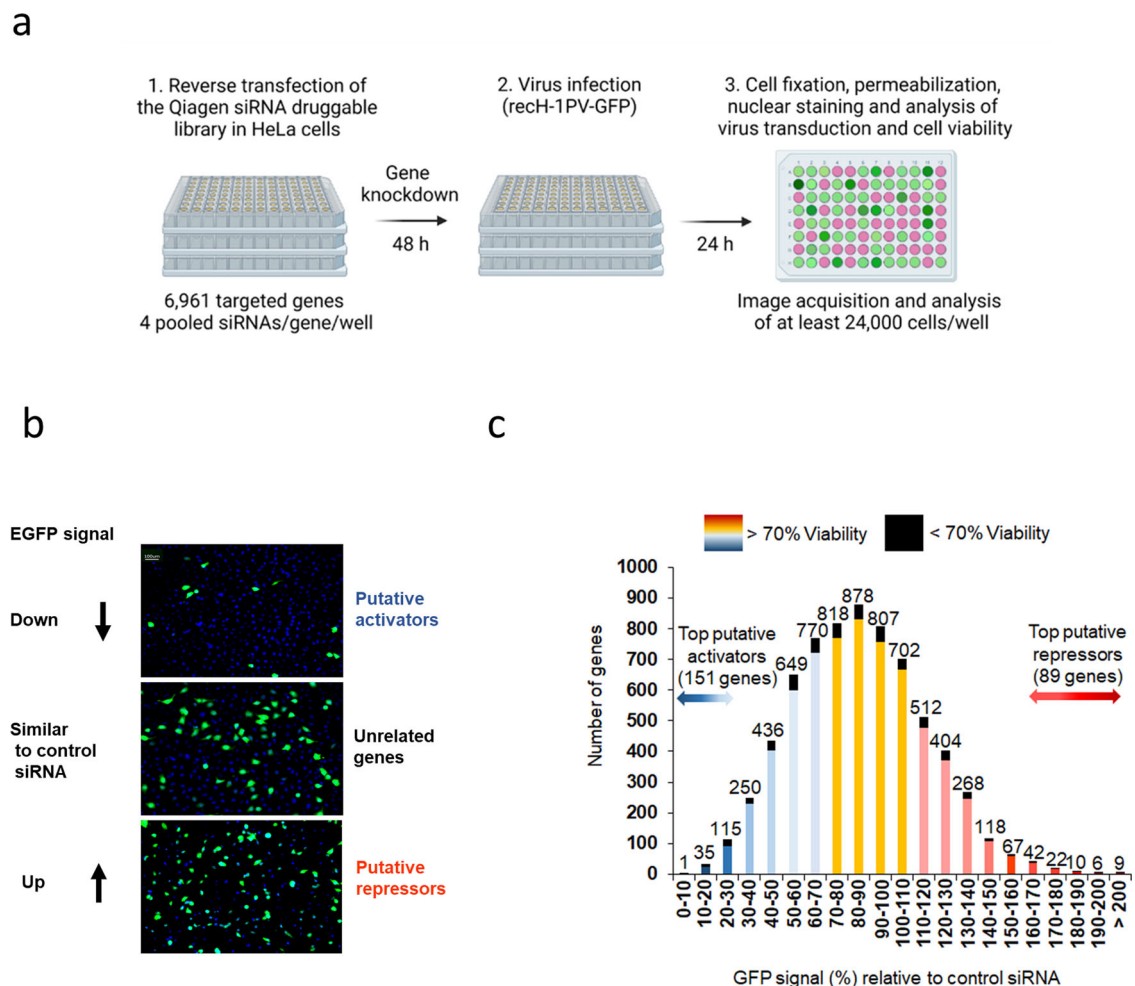

**Fig. 1 siRNA library screening reveals putative modulators of H-1PV life cycle. a** Protocol design. The complete siRNA human druggable genome library (Qiagen), consisting of 6961 siRNAs pools (four siRNAs/pool/gene), was spotted onto 96-well plates (one pool per well) in triplicate and then reverse transfected in HeLa cells. After 48 h, cells were infected with recombinant H-1PV harbouring the EGFP reporter gene (recH-1PV-EGFP). The EGFP signal was quantified 24 h after infection as a measure of H-1PV transduction efficacy. Internal positive and negative controls were added to each plate to check inter-plate and day-to-day variability, as described in the Materials and Methods.This figure panel was created using BioRender.com. **b** Representative fluorescence images. The concentration of recH-1PV-EGFP that converted 50% of scramble siRNA-transfected cells to EGFP-positive was used for the infection. The acquired EGFP signal was used as a baseline for normalisation of virus transduction obtained in cells transfected with siRNAs targeting every gene present in the siRNA library. The scale bar depicted in the figure corresponds to 100 μm. **c** Schematic distribution of siRNA library screening results. The x axis in the graph represents the percentage of GFP signal relative to control siRNA. H-1PV activators are depicted in shades of blue, H-1PV repressors in shades of red, and genes whose silencing did not affect H-1PV transduction in yellow. Each group is further differentiated based on the percentage of cell viability, which was determined by silencing particular genes in non-infected cells transfected with siRNA pools and testing the effect on cell viability: <70% viability (black, these genes were not further analysed) and >70% viability (coloured, selected genes). Numbers on top of the columns indicate the number of genes present in each group.

significantly, so they are less likely to be implicated in the H-1PV life cycle. Genes whose silencing in the absence of virus killed >70% of cells were not analysed further, because they were considered essential for cell viability (Fig. 1c).

H-1PV cell surface binding and entry is SA-dependent[27]. Neuraminidase (NA) treatment, which cleaves SA groups from glycoproteins, dramatically impairs H-1PV infectivity[27]. The siRNA library used for the screening included glucosamine (UDP-N-acetyl)-2-epimerase/N-acetylmannosamine kinase (GNE) as the only representative gene involved in SA biosynthesis (other genes involved in SA metabolism were not present in the library). In agreement with a role for SA in H-1PV entry, siRNA-mediated knockdown of GNE decreased H-1PV transduction by 69.64%. For the present study, the top 151 putative activators were run on the Ingenuity Knowledge Base to analyse the possible role of these putative activators in canonical cellular pathways (Supplementary

Figure 1a). Ingenuity-pathway analysis revealed the previously known association of H-1PV with canonical pathways such as Toll-like receptor signalling[34] and NF-kB signalling[34], as well as other new possible pathways that could modulate the H-1PV life cycle, including protein kinase A signalling, eukaryotic initiation factor 2 signalling and the STAT3 pathway, among others.

We then carried out gene ontology analysis on the putative activators, which were classified based on their subcellular localisation. Most of these activators was predicted to reside in the cytoplasm (46%), followed by the plasma membrane (22%), nucleus (20%) and extracellular space (7%) (Supplementary Figure 1b). To gain insight into their cellular function, we grouped activators based on PANTHER Protein Class. They belong to a wide spectrum of protein classes involved in different cellular processes (Supplementary Figure 1c). The top five protein classes were protein modifying enzyme (25.30%),

metabolite interconversion enzyme (22.20%), translational protein (7.10%), transmembrane signal receptor (7.10%) and gene-specific transcriptional regulator (6.10%). As our goal was to identify factors required in cell attachment and entry, we focused on two protein classes: ECM protein (2%, two genes) and transmembrane signal receptor (7.1%, seven genes). The ECM protein class included laminin γ1 chain and galectin-1. Silencing of the genes encoding these two proteins, namely *LAMC1* and *LGALS1*, reduced H-1PV transduction by >70% (Supplementary Table 1a). The transmembrane signal receptor protein class included cholecystokinin A receptor, TGF-beta receptor type-2, Serine/threonine-protein kinase receptor R3, D (4) dopamine receptor, melanocortin receptor 4, interferon alpha/beta receptor 2, and lysophosphatidic acid receptor 5. Silencing of the genes encoding these seven proteins reduced H-1PV transduction by >70% (for *ACVRL1*, *DRD4*, *MC4R*, *IFNAR2*, and *GPR92*) or >80% (for *CCKAR* and *TGFBR2*) (Supplementary Table 1b). However, after consulting the Human Protein Atlas (https://www.proteinatlas.org/), we found that only four genes, namely *LAMC1*, *LGALS1*, *TGFBR2* and *IFNAR2*, are expressed in HeLa. In particular, *LAMC1* attracted our attention. This gene encodes the laminin γ1 chain, a member of the laminin family. Laminins are known to be heavily glycosylated proteins that are rich in terminal SA residues. Glycan chains bearing terminal SA contribute significantly to laminin adhesion to the cell surface. Given the dependency of H-1PV infection on SA, in the present study, we decided to investigate whether laminins mediate H-1PV cell attachment.

**H-1PV uses laminin γ1 for its attachment at the cell surface**. To test the biological role of laminin γ1 in H-1PV cell attachment and entry, we first silenced *LAMC1* expression in HeLa cells using two siRNAs targeting two distinct regions of the gene and then assessed the ability of H-1PV to bind to and penetrate these cells at 4 °C (virus binding assay) and 37 °C (virus cell uptake assay). We observed a marked decrease in virus cell binding and virus cell uptake in cells transfected with *LAMC1* siRNAs in comparison with control siRNA (Fig. 2a, b). HeLa cells transfected with siRNAs, as above, were also used for a virus transduction assay in which cells were infected with recH-1PV-EGFP, the same recombinant virus previously used for the siRNA library screening. In agreement with the results obtained from the siRNA library screening in which a *LAMC1* siRNA pool was used, individual *LAMC1* siRNAs strongly decreased H-1PV transduction (Fig. 2c). We next evaluated the impact of siRNA-specific *LAMC1* knockdown on H-1PV-induced cytotoxicity by measuring cell viability in siRNA-transfected HeLa cells. As expected, untransfected cells or siRNA control transfected cells were efficiently killed by H-1PV (77.2% and 71.7% reduction of cell viability, respectively). By contrast, HeLa cells transfected with *LAMC1* siRNA were significantly more resistant to H-1PV-induced cytotoxicity (Fig. 2d). We then performed an antibody blocking experiment using specific anti-LAMC1 antibody to block H-1PV cell entry. HeLa cells preincubated with anti-laminin γ1 antibody were significantly less susceptible to H-1PV infection than cells preincubated with control IgG isotype (Fig. 2e). We also generated a stable *LAMC1* KD HeLa cell line in which the *LAMC1* gene was knocked down via CRISPR-Cas9 genome editing technology (our initial goal was to completely knock out the *LAMC1* gene, but this was not possible in our hands despite several attempts). A significant decrease in H-1PV infection was observed in *LAMC1* KD cell line compared with parental HeLa cells. This reduction of H-1PV infectivity was rescued by transient transfection of the *LAMC1* gene in these cells

(Fig. 2f). In agreement with these results, overexpression of *LAMC1* in parental HeLa cells significantly increased H-1PV cellular uptake (Fig. 2g) and H-1PV transduction (Fig. 2h).

**Laminins are involved in H-1PV cell attachment and entry**. Laminins chains are assembled through disulphide bonds to form a trimeric cruciform structure consisting of one long and three short arms. Five α chains (α1–5), four β chains (β1–4) and three γ chains (γ1–3) have been described to date, accounting for the 16 known laminin isoforms. Laminin molecules are named according to their chain composition, e.g., laminin 111 contains the α1, β1 and γ1 chains.

To verify whether H-1PV interacts with laminin(s) at the cell surface, we first carried out colocalization studies in which NCH125 glioma cells were infected with Alexa Fluor 488-labelled virus. The infection was conducted on ice to block virus internalisation. Cells were fixed and laminins were detected using a pan-laminin antibody. Confocal analysis showed that a consistent fraction of the virus was found to be associated with laminins (Fig. 3a).

To determine which other laminin chains in addition to laminin γ1 are involved in H-1PV cell binding, we carried out competition experiments using a panel of commercially available purified soluble trimeric laminins, namely laminins 111, 121, 211, 221, 411, 421, 511 and 521, which all contain the γ1 chain, and laminin 332. We used soluble fibronectin and collagen, two other common constituents of the ECM, as controls. Cells were preincubated with laminins, fibronectin or collagen before being infected with the recH-1PV-EGFP virus. Preincubation with fibronectin or collagen had no significant effect on H-1PV transduction. By contrast, a significant decrease in virus transduction was observed in cells pre-treated with all laminins containing the γ1 chain, but not with laminin 332 (Fig. 3b and Supplementary Figure 2). These results provide evidence that the γ1 chain is required for H-1PV binding to laminins and that different laminin polypeptides containing the γ1 chain may be involved in virus cell binding and entry.

By interrogating the Human Protein Atlas, we found that HeLa cells express seven genes encoding laminin chains (*LAMA1*, *LAMA4*, *LAMA5*, *LAMB1*, *LAMB2*, *LAMB3* and *LAMC1*), whereas five genes of the same family (*LAMA2*, *LAMA3*, *LAMB4*, *LAMC2* and *LAMC3*) were below detection levels (Supplementary Figure 3a).

To confirm the involvement of other laminins in H-1PV cell attachment, we silenced the expression of another member of the laminin gene family, *LAMB1*, which encodes the laminin β1 chain. The choice of *LAMB1* was suggested by the fact that its silencing in our siRNA library screening was associated with a strong decrease in H-1PV transduction efficiency (~60% less than that obtained with control siRNA), which positioned it as the second-most active gene encoding laminin chains after *LAMC1* (Supplementary Figure 3b). Similarly, to the results obtained with *LAMC1*, silencing of *LAMB1* using two different siRNAs targeting two distinct regions of the *LAMB1* gene strongly decreased H-1PV cell binding and entry, transduction and oncolysis (Supplementary Figure 4).

We then assessed the role of *LAMC1* and *LAMB1* in H-1PV cellular uptake in two other cancer cell lines, namely HCT116 (colorectal carcinoma) and A549 (lung adenocarcinoma). As shown for HeLa cells (Fig. 2), specific silencing of *LAMC1* and *LAMB1* significantly decreased H-1PV cellular uptake in both cell lines compared to control siRNA (Supplementary Figure 5).

**Pre-treatment with heparin blocks H-1PV infection**. Laminins contain several binding sites for heparin[35,36]. We hypothesised

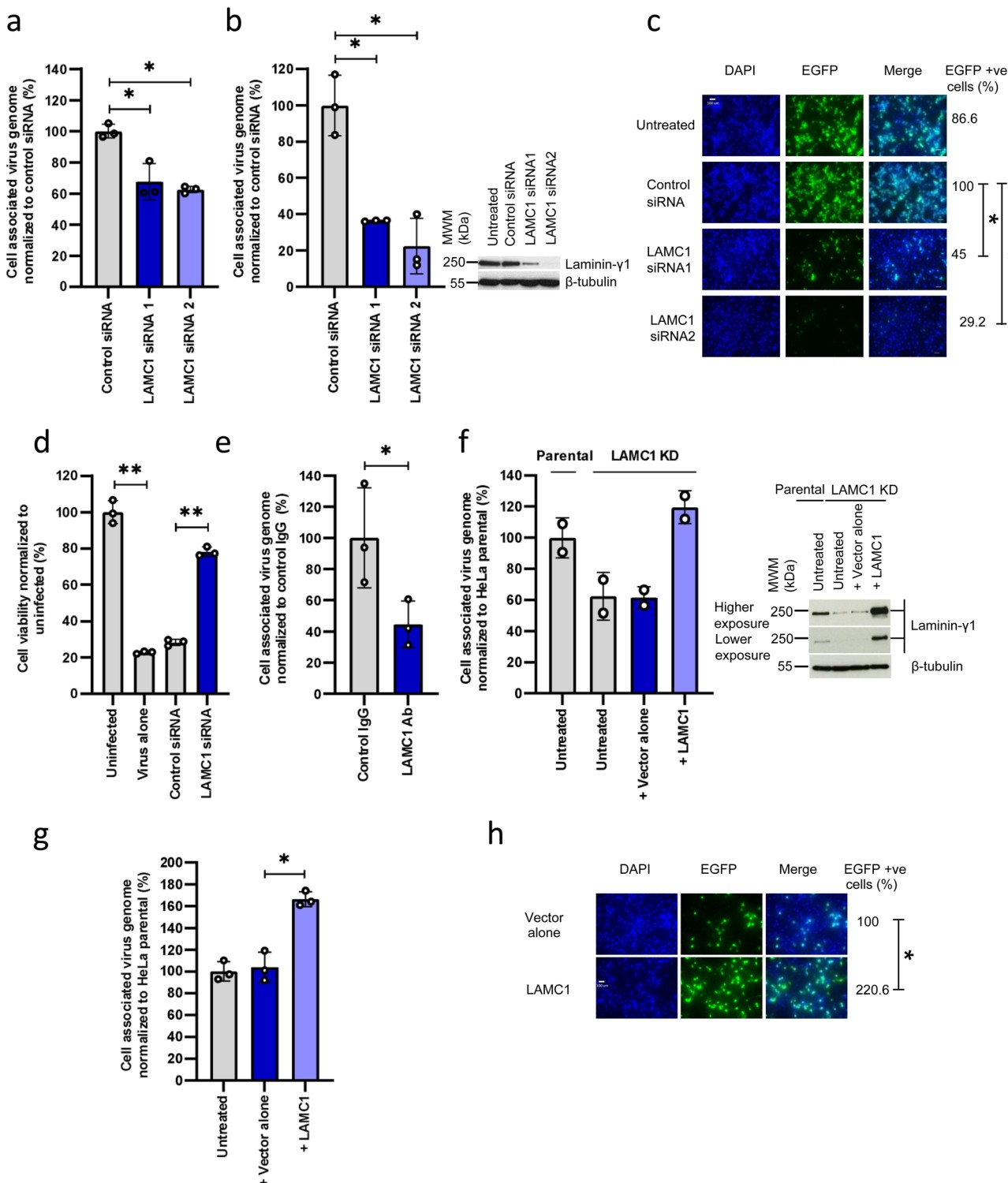

that treatment with heparin may interfere with H-1PV infectivity by competing with the virus for binding to laminin(s). To test this hypothesis, HeLa cells were pre-treated with different concentrations of soluble heparin before being infected with H-1PV. NA was used as a positive control for its ability to prevent H-1PV cell binding and entry by cleaving surface SA. Furthermore, we pre-treated cells with heparinase III, an enzyme known to degrade heparan sulphate glycosaminoglycan chains, which are abundant components of the cell surface and ECM. As expected, treatment with NA strongly inhibited H-1PV infectivity,

confirming the importance of SA for H-1PV entry. Incubation with heparin, but not heparinase III, also decreased the ability of H-1PV to penetrate cells (Fig. 4a). This effect was proportional to the concentration of heparin used (Fig. 4b). Consistent with this observation, heparin treatment reduced H-1PV transduction efficacy (Fig. 4c) and protected cells from H-1PV-induced oncolysis (Fig. 4d, e). Treatment with a peptide resembling the heparin-binding sites, which is present in the laminin globular domains at the C-terminus of the laminin α chains, also decreased H-1PV transduction (Fig. 4f).

**Fig. 2 Laminin γ1 is involved in H-1PV cell attachment and entry. a** Silencing of *LAMC1* decreases H-1PV cell binding. HeLa cells were transfected with control siRNA or two distinct siRNAs targeting two separate regions of *LAMC1*. At 46 h post transfection, cells were infected with H-1PV (MOI 1 pfu/cell) for 2 h at 4°C. The results are presented as a percentage of cell-associated virus genome normalised to control siRNA. The representative experiment shown in the figure is repeated twice; *n* = 3 biologically independent samples; control siRNA vs *LAMC1* siRNA1 (*P* = 0.0165) and control siRNA vs *LAMC1* siRNA2 (*P* = 0.0101). **b** Silencing of *LAMC1* decreases H-1PV cell uptake. HeLa cells were transfected with the same siRNAs used for **a**. At 46 h post transfection, cells were infected with H-1PV (MOI 1 pfu/cell) for 4 h at 37°C. Cell processing and data representation were performed as per **a**. The independent experiment shown is repeated twice; *n* = 3; Control siRNA vs *LAMC1* siRNA1 (*P* = 0.0231) and control siRNA vs *LAMC1* siRNA2 (*P* = 0.0498). Western blot analysis ascertained the downregulation of *LAMC1* in siRNA-transfected cells using β-tubulin as a loading control. **c** Silencing of *LAMC1* decreases H-1PV transduction. HeLa cells were transfected with the same siRNAs used for **a**. At 46 h post transfection, cells were infected with recH-1PV-EGFP (1 TU/cell) and grown for an additional 24 h and analysed for EGFP-positive (+ve) cells. Numbers indicate the fraction of EGFP-positive cells (%) normalised to that obtained in cells transfected with control siRNA. The scale bar depicted in the figure corresponds to 100 μm. **d** Silencing of *LAMC1* protects cells from H-1PV oncotoxicity. HeLa cells were transfected with control or LAMC1 siRNAs. At 72 h post transfection, cells were infected or not with H-1PV (MOI 0.25 pfu/cell) and grown for an additional 72 h before measuring cell viability by the CellTitre-Glo 2.0 assay. The data shown represent the percentage of cell viability normalised to uninfected cells. The independent experiment shown is repeated twice; *n* = 3 biologically independent samples; uninfected vs virus alone (*P* = 0.0027); Control siRNA vs *LAMC1* siRNA (*P* = 0.0024). **e** Anti-laminin γ1 antibody impairs H-1PV cell uptake. HeLa cells were incubated with either IgG isotype (control) or anti-laminin-γ1 chain antibodies for 45 min on ice. An H-1PV cell membrane binding/entry assay was performed by treating the cells with H-1PV (MOI 0.1 pfu/cell) first for 30 min on ice and then for 60 min at 37°C in a serum-free medium. Cells were then processed as described in **a**. The independent experiment shown is repeated twice; *n* = 3 biologically independent samples; control IgG vs LAMC1 Ab (*P* = 0.031). **f** CRISPR-Cas9-mediated knockdown of *LAMC1* impairs H-1PV uptake, which is rescued by re-introduction of exogenous *LAMC1*. Parental HeLa and LAMC1 KD (a *LAMC1*-knockdown HeLa cell line constructed using CRISPR-Cas9 technology) were transfected or not with either empty vector (+ vector alone) or vector expressing *LAMC1* (+LAMC1). At 48 h post transfection, cells were infected with H-1PV (MOI 100 pfu/cell) for 4 h at 37°C for the virus uptake assay. Cells were then processed as described in **a**. The independent experiment shown is repeated twice; *n* = 2 biologically independent samples. **g** Overexpression of *LAMC1* enhances H-1PV uptake. HeLa cells were transfected either with empty vector (+ vector alone) or vector carrying *LAMC1*. Cells were then processed as described in **a**. The independent experiment shown is repeated twice; *n* = 3 biologically independent samples; + vector alone vs + LAMC1 (*P* = 0.0329). **h** Overexpression of *LAMC1* in HeLa enhances H-1PV transduction. HeLa cells were transfected with either vector alone or vector expressing *LAMC1*. At 46 h post transfection, cells were infected with recH-1PV-EGFP (0.25 TU/cell) and grown for an additional 24 h and processed as described above. Scale bar corresponds to 100 μm. Statistical significance was calculated using a paired two-tailed *t* test by GraphPad Prism 8; \**P* < 0.05; \*\**P* < 0.01. Error bars for all data indicate the mean values ±SD. Source data are provided as Source Data file.

Inhibition of virus binding/entry and transduction was also confirmed in NCH125 glioma-derived and BxPC3 pancreatic carcinoma-derived cell lines upon heparin and NA treatment (Supplementary Figure 6a, b). Together, these results provide strong evidence that laminins have an essential role in H-1PV infectivity at the level of virus cell attachment and entry.

**H-1PV directly binds to laminins through SA.** To investigate whether H-1PV can directly bind to laminins without requiring other factors, we performed enzyme-linked immunosorbent assay (ELISA) experiments in which H-1PV was added to microplate wells pre-coated with purified laminins (namely laminins 211, 221, 411, 421 and 332). As controls, wells were also pre-coated with collagen IV, fibronectin or bovine serum albumin (BSA). Stronger binding of H-1PV was obtained in wells pre-coated with laminins containing the γ1 chain, while binding to the wells pre-coated with laminin 332, collagen IV and fibronectin was much less efficient and closer to that seen for BSA (Fig. 5a). To investigate whether SA mediates H-1PV binding to laminins, we repeated the ELISA by treating laminin-pre-coated wells with NA, to remove SA residues. In agreement with cell culture experiments, treatment with NA almost completely abolished the H-1PV-laminin interactions (Fig. 5b).

Finally, we repeated in this setting the competition experiment with heparin. Laminin-pre-coated wells were pre-treated with heparin before the addition of H-1PV. Pre-treatment with heparin strongly reduced H-1PV-laminin interactions (Fig. 5c). Control ELISA experiments in which wells were pre-coated with BSA-conjugated heparin excluded the possibility that the virus was binding to heparin, as H-1PV-binding efficiency in these wells was similar to that obtained in wells pre-coated with BSA alone (background) (Fig. 5c). Together, these results provide further evidence that H-1PV binds to laminins via SA moieties

presumably present within the heparin-binding site(s) and that sialylated laminins mediate H-1PV attachment at the cell surface.

***LAMC1* upregulation is associated with poor prognosis in gliomas and other tumours.** Bioinformatic analysis using the gene expression profiling interactive analysis tool showed that *LAMC1* is differentially expressed (down- and upregulated in comparison with paired normal tissues) across the various tumours (Supplementary Figure 7). For example, *LAMC1* upregulation was found in pancreatic carcinoma and GBM, the two tumour entities in which H-1PV has been tested in the clinic to date. By analysing data from The Cancer Genome Atlas (TCGA, http://cancergenome.nih.gov) with Cox regression models, we found that *LAMC1* and *LAMB1* are statistically associated with poor patient survival in different cancers (Supplementary Fig. 8). In gliomas, *LAMC1* expression increases with grade, where *LAMC1* is more expressed in GBM (grade IV) than lower grade gliomas (Supplementary Fig. 9). Concomitantly, high *LAMC1* expression is associated with poor prognosis in gliomas and in GBM according to data from the Gravendeel, Rembrandt and TCGA data sets (Supplementary Fig. 10).

**Laminin γ1 chain is differentially expressed in primary and recurrent GBM biopsies.** Next, we analysed the expression pattern of laminin γ1 in human GBM biopsies to assess whether the protein is expressed ubiquitously or in only a certain fraction of tumour specimens. To this end, we used a 'home-made' GBM Tissue microarray, which includes 110 different GBM patient biopsies obtained from 61 primary and 49 recurrent GBMs, and 12 normal tissues (from brain, liver and tonsil), and performed immunohistochemical staining using an anti-laminin γ1 chain antibody. A significant difference in the protein levels of laminin γ1 chain was observed between normal tissues and GBMs.

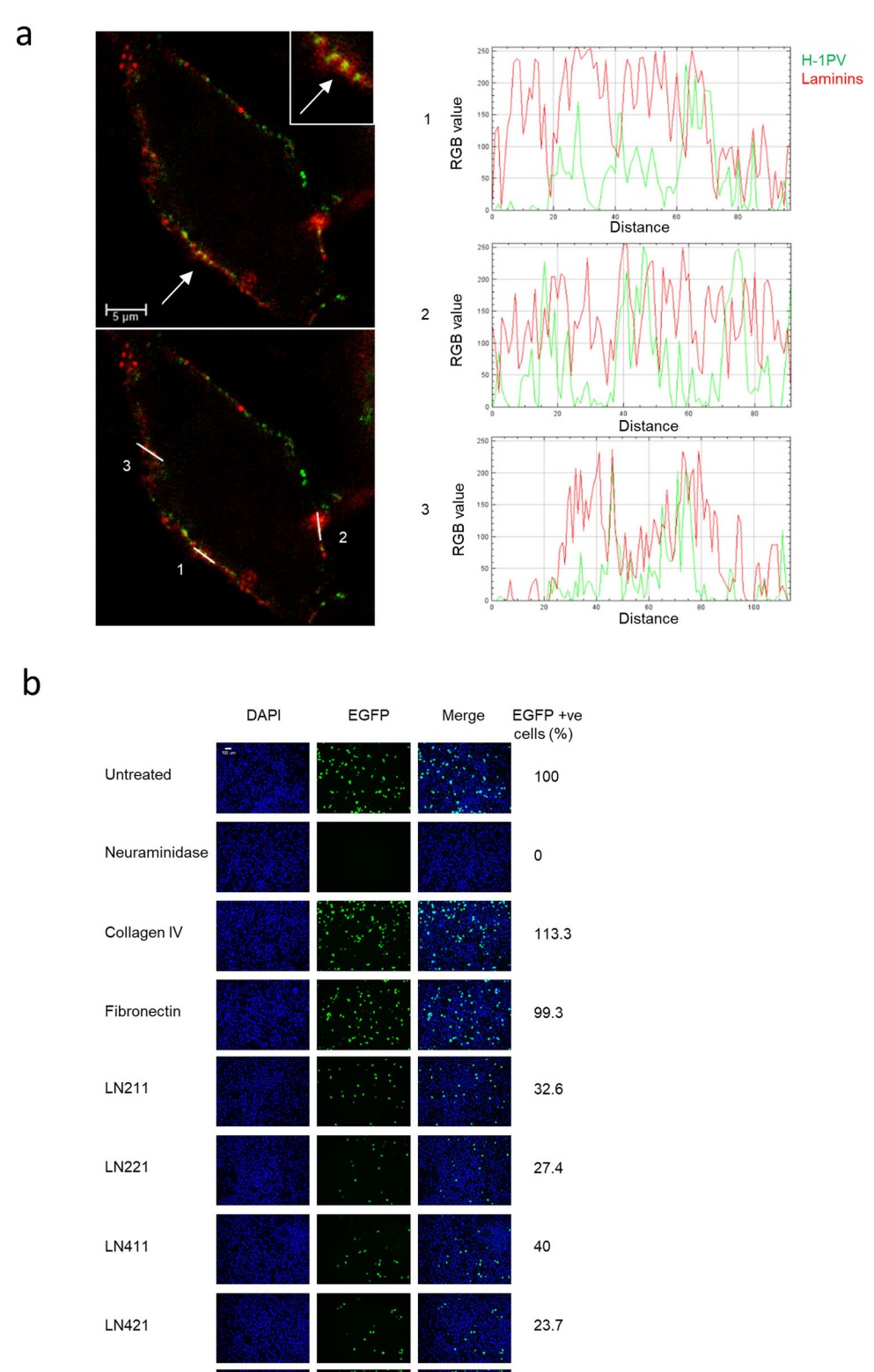

Moreover, while low in all normal tissues, the protein levels of laminin γ1 varied across the GBMs biopsies, with a fraction of biopsies expressing high (>40% of positive cells), medium (20–40% of positive cells) or low (<20% of positive cells) levels. Comparison of laminin γ1 staining between primary GBM and recurrent GBM biopsies did not show any significant statistical difference. However, the highest levels of laminin γ1 chain (>20%

of positive cells) were observed in recurrent GBM biopsies (Supplementary Fig. 11).

**LAMC1 expression positively correlates with H-1PV oncolysis.** We hypothesised that *LAMC1* is one of the critical factors for a successful H-1PV infection. To verify this hypothesis, we screened

**Fig. 3 Laminins are the cell attachment factors of H-1PV. a** NCH125 glioma cells were placed on ice and then infected with Alexa Fluor 488-labelled H-1PV (~MOI 50 pfu/cell) for 2 h. Cells were then fixed, and stained with pan-laminin antibody. Images were acquired using a ×63 objective with the Leica TCS SP5 II confocal microscope. Merged image show colocalization of H-1PV (green channel) and laminin (red channel). An enlargement of a portion of the image is also shown (arrow). Colocalization analysis was performed with ImageJ RGB-plus profiler plugin. Scale bar = 5 μm. **b** Treatment with soluble laminins impairs H-1PV cell transduction. HeLa cells were preincubated with 2 μg/ml of indicated laminins or collagen IV or fibronectin for 24 h and then infected with recH-1PV-EGFP (0.5 TU/cell) for an additional 27 h. Neuraminidase (NA) [0.1 U/ml] was used as a positive control for blocking virus infection. Cells were then processed as described in Fig. 2c. Images were acquired using a ×10 objective with the BZ-9000 fluorescence microscope (Keyence). Scale bar = 100 μm.

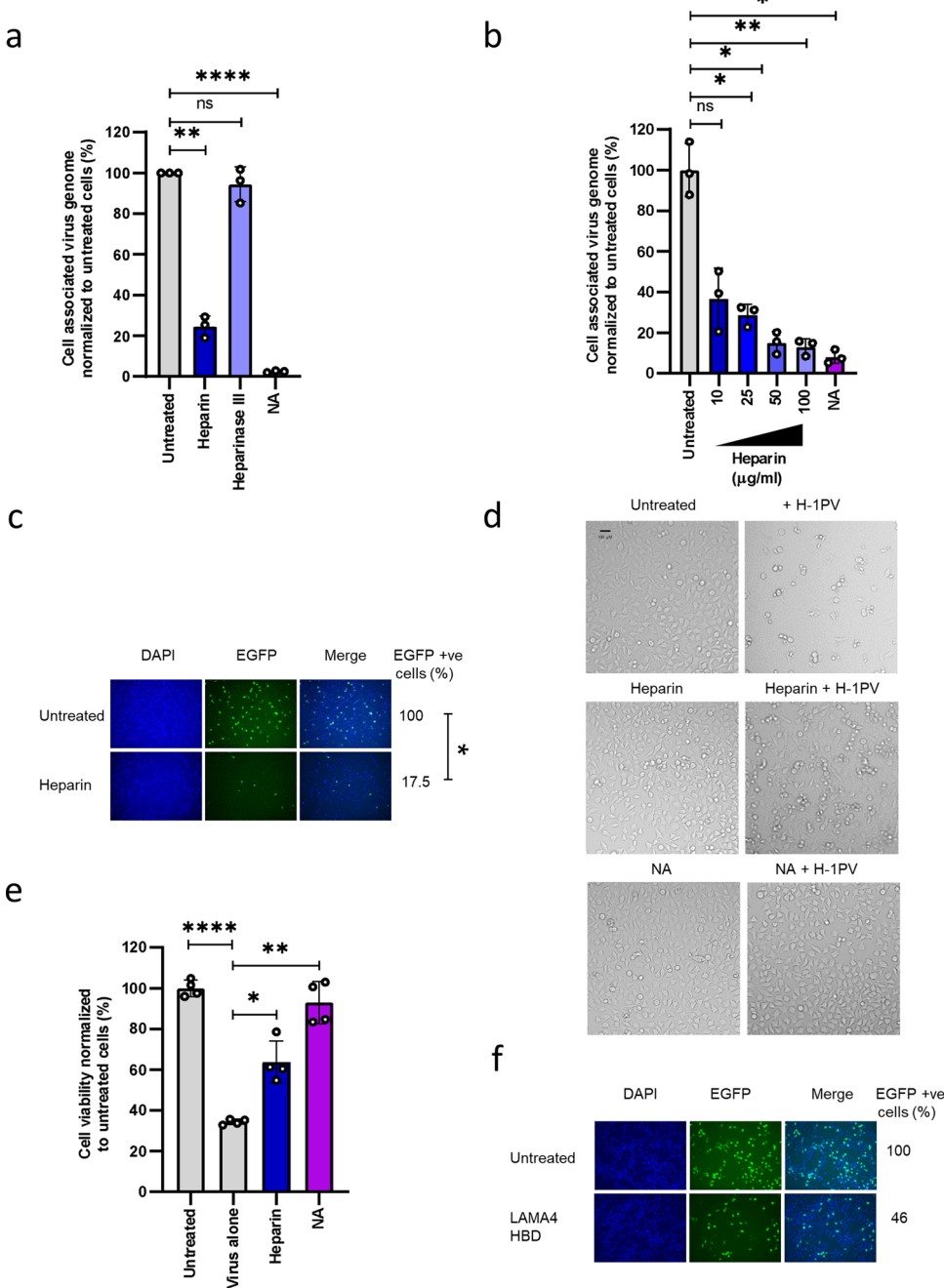

53 cancer cell lines from the NCI-60 cancer cell line panel for their sensitivity to H-1PV-induced oncolysis. We checked whether a direct correlation could be established between susceptibility to virus infection and *LAMC1* gene expression levels. The NCI-60 panel is composed of cancer cell lines derived from different tumour types, namely lung, central nervous system (CNS), melanoma, breast, renal, ovarian, colon, prostate and leukaemia, which are fully characterised in regard to their gene expression profile and drug sensitivity. Screening was performed using the xCELLigence system, which allows real-time monitoring of cell proliferation. Typical examples of H-1PV-sensitive (SNB-75) or H-1PV-resistant (COLO205) cancer cell lines are shown in Fig. 6a.

**Fig. 4 Heparin treatment inhibits H-1PV binding to laminins. a** Treatment with heparin, but not removal of cell surface heparan sulphate by heparinase III, impairs H-1PV uptake. HeLa cells were pre-treated with heparin, heparinase III or neuraminidase (NA) and then infected with H-1PV (MOI 1 pfu/cell) for 4 h at 37°C. Cells were then processed as described in Fig. 2 in order to isolate and quantify the fraction of H-1PV particles associated with the cells. The independent experiment shown is repeated thrice; $n = 3$ biologically independent samples; untreated vs heparin ($P = 0.0016$); untreated vs heparinise III ($P = 0.3723$); untreated vs NA ($P < 0.0001$). **b** Treatment with heparin decreases H-1PV cell binding and entry in a dose-dependent manner. HeLa cells were preincubated with the indicated concentrations of heparin and then infected with H-1PV for 4 h at 37°C. The independent experiment shown is repeated twice; $n = 3$ biologically independent samples. Untreated vs heparin-10 μg/ml ($P =$ ns); untreated vs heparin-25 μg/ml ($P = 0.0205$); untreated vs heparin-50 μg/ml ($P = 0.0156$); untreated vs heparin-100 μg/ml ($P = 0.0043$); untreated vs NA ($P = 0.0104$). **c** Treatment with heparin decreases H-1PV transduction. HeLa cells were preincubated with heparin (100 μg/ml) for 18 h and then infected with recH-1PV-EGFP (0.5 TU/cell) for an additional 24 h. Cells were then processed as described in Fig. 2c. **d** Heparin protects HeLa cells from H-1PV oncolytic activity. A higher fraction of HeLa cells survived to H-1PV infection (96 h) when pre-treated with heparin compared with untreated cells. NA was used as a positive control for blocking virus cell binding/ entry and thereby preventing H-1PV-mediated oncolytic activity. **e** Treatment with heparin protects cells from H-1PV oncotoxicity. HeLa cells were pre-treated with heparin (100 μg/ml) or neuraminidase (0.1 U/ml) for 18 h and then infected with H-1PV (MOI 1 pfu/cell) for 96 h. Cell viability was assessed by the CellTitre-Glo 2.0 assay. The results are presented as the percentage of cell viability normalised to untreated cells. The independent experiment shown is repeated twice; $n = 3$ biologically independent samples; Untreated vs virus alone ($P < 0.0001$); untreated vs heparin ($P = 0.0122$); untreated vs NA ($P = 0.0013$). **f** Treatment with a peptide corresponding to the heparin-binding domain (HBD) present within the laminin G domain-like modules of laminin α 4 (LAMA4) impairs H-1PV cell transduction. The reduction in transduction activity observed upon peptide treatment suggests a physical interaction between the peptide and H-1PV capsid that impairs virus infectivity. Statistical significance was calculated using a paired two-tailed $t$ test by GraphPad Prism 8 with *$P < 0.05$; **$P < 0.01$; ***$P < 0.001$; ****$P < 0.0001$; ns: not significant. Error bars for all data indicate the mean values ± SD. Source data are provided as Source Data file.

The EC50 was then calculated at 24, 48, 72 and 96 h post infection (Supplementary Fig. 12). We found 36 cancer cell lines of various origin to be highly susceptible to H-1PV infectivity (cytostatic and cytotoxic effects were observed at MOI ≤ 10); 11 cancer cell lines to have low sensitivity (effects observed only at MOI ≥ 10–50); and 6 cancer cell lines to be resistant to the highest H-1PV concentration used (MOI 50) (Fig. 6b and Supplementary Fig. 12). Cancer cell lines derived from lung, CNS and breast cancers and melanoma were among the most sensitive to H-1PV infection. The only exceptions were LOX IMVI (melanoma) and MCF7 (breast cancer), which were refractory to the highest viral concentration used. By contrast, cell lines derived from colon and ovarian cancers were among the most resistant to H-1PV infection, with HCT-15, HCC-2998, COLO205 (colon carcinomas) and OVCAR-3 (ovarian cancer) resistant to the highest viral concentration used.

Next, we investigated the correlation between *LAMC1* expression and H-1PV oncotoxicity using EC50 values and expression data from the NCI-60 panel (Fig. 6c). In the 53 cell lines considered here, *LAMC1* expression is robustly anticorrelated with EC50 values (Pearson correlation, $r = -0.52$, $P = 6.2E-05$) (Fig. 6c, d). To verify this observation, we repeated the correlation analysis using an independent gene expression data set (CCLE data set), and observed highly consistent results ($r = -0.52$, $P = 7.9E-04$, Fig. 6d). We also found that the gene expression of the other laminins is anticorrelated with their EC50, in particular: *LAMA1* ($r = -0.32$, $P = 1.76E-2$), *LAMA4* ($r = -0.28$, $P = 4.11E-2$) and *LAMB2* ($r = -0.39$, $P = 4.36E-3$) (Supplementary Fig. 13). Although such correlations are weaker than that observed in the case of *LAMC1*, these results provide additional evidence of the relevance of laminins for H-1PV oncolysis.

***LAMC1* expression positively correlates with H-1PV oncolysis in glioma cell lines and 3D organotypic spheroid cultures.** Based on the results shown above, we further analysed six glioma-derived cell lines for their sensitivity to H-1PV oncolysis. As a control, normal human astrocytes were used. Cells were infected for 72 h with H-1PV at MOI 10. As expected from previous published results[19,37,38], normal astrocyte were quite resistant to H-1PV infection. In two cell lines, NCH125 and NCH37, >60% of cells were killed by the virus. By contrast, U251, LN308, T98G and A172-MG were less susceptible to H-1PV oncolysis; <30% of cells in each line were killed (Fig. 6e). Total RNA was isolated

from these cell cultures and subjected to NanoString analysis to quantify *LAMC1* expression levels. In agreement with the NCI-60 cell line screening, a positive correlation between *LAMC1* mRNA levels and H-1PV oncolysis was also found in glioma cell lines (Fig. 6e).

Next, we analysed two GBM primary 3D organotypic spheroid cultures derived from GBM patient-derived orthotopic xenografts, as previous gene expression analysis indicated low (P3) or high (P13) *LAMC1* mRNA levels in these tumours[39]. Spheroids were infected with the recombinant H-1PV expressing EGFP for 5 days. NanoString analysis confirmed differential *LAMC1* expression in the GBM spheroids (Fig. 6f). In agreement with previous results, H-1PV infectivity was much more efficient in P13 spheroids expressing high levels of *LAMC1* than low-*LAMC1* P3 cells (Fig. 6f). Together, these results further validate the importance of *LAMC1* for the H-1PV life cycle and advance the possibility that its expression may be used to predict the success of H-1PV infection.

**Discussion**
To identify host cell factors mediating H-1PV cell surface binding and entry, we performed a druggable genome-wide siRNA library screening. We identified 151 activators and 89 repressors affecting H-1PV transduction by >70%, plus a number of other genes modulating H-1PV transduction to a lesser degree (Fig. 1). These factors could directly or indirectly modulate H-1PV transduction in a positive or negative manner by participating in various steps of the virus life cycle, from early events such as cell attachment and entry, cytoplasm trafficking and nuclear entry, to later events such as P4 early promoter activation, which leads to *NS* gene unit expression and subsequent NS1-mediated activation of the P38 late promoter.

It has been described that SA is required for H-1PV binding and entry[27]. In this study, we also confirmed that cleavage of SA by NA treatment, completely abolished H-1PV cell binding and transduction. Consistently, siRNA-mediated silencing of *GNE*, the only gene involved in SA biosynthesis present in the siRNA library used for the screening, dramatically reduced H-1PV transduction.

We focused on candidates that literature searches and bioinformatic tools predicted to be part of the ECM. We selected *LAMC1*, encoding the laminin γ1 chain, because laminins have been previously described as interacting with SA. We demonstrated

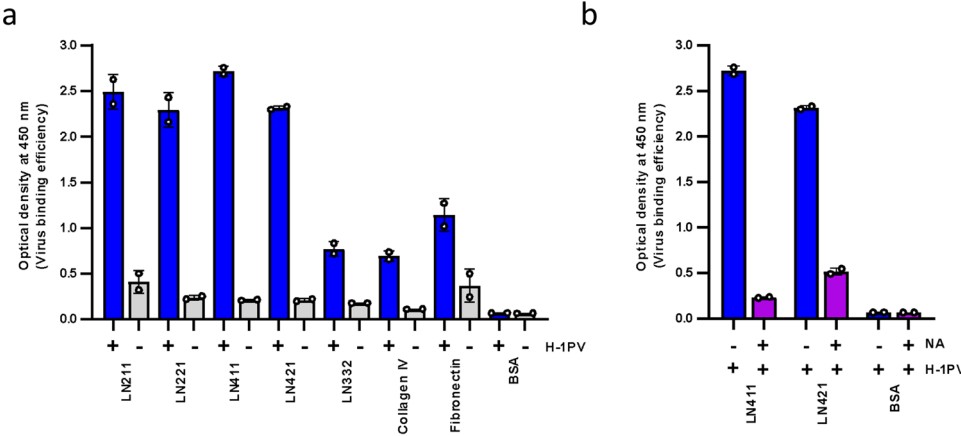

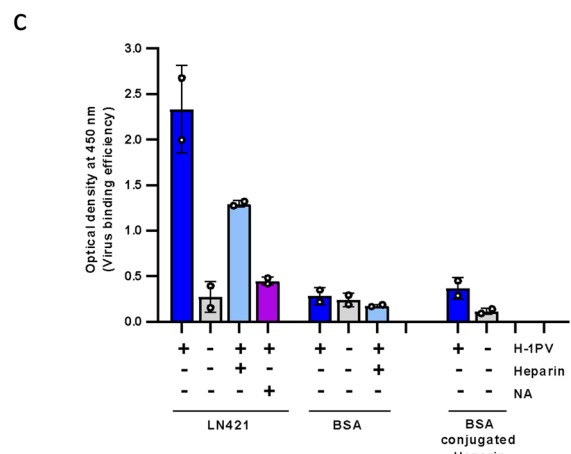

**Fig. 5 H-1PV directly binds to laminins through sialic acid presumably within the heparin-binding site. a** H-1PV binds to laminins in ELISA. Selected wells of a 96-well microtiter plate were pre-coated with purified LN211, LN221, LN411, LN421, LN332, collagen type IV, fibronectin and BSA. After blocking non-specific binding, wells were treated with H-1PV or left untreated, followed by stringent washing to remove loosely bound virus particles. The bound virus particles were detected by ELISA using a virus capsid conformational antibody as described in the M&M. The independent experiment shown is repeated twice; $n = 2$. **b** H-1PV binds to laminin via sialic acid moieties. Wells pre-coated with purified LN411, LN421 or BSA were treated or not with NA. The bound virus particles were detected by ELISA as described in Fig. 5a. The independent experiment shown is repeated twice; $n = 2$. **c** Treatment with heparin decreases H-1PV-interaction with laminins without binding to the virus capsid. Selected wells of a 96-well microtiter plate were pre-coated with purified LN421, BSA-conjugated heparin or BSA. The wells with applied LN421 or BSA were pre-treated with heparin or NA or left untreated. ELISA was performed as above. Each column represents the mean value ± the standard deviation ($n = 2$). The independent experiment shown is repeated twice; $n = 2$. Error bars for all data indicate the mean values ± SD. Source data are provided as Source Data file.

that laminins play a key role in virus cell attachment and entry and that the interaction is mediated by the SA moieties present in these ECM proteins (Fig. 7). Thus, laminins, by displaying SA, seem to act as primary cell attachment factors for H-1PV. However, most likely other glycoproteins (or other SA-bearing molecules) may participate in H-1PV cell attachment and entry. This is also supported by our results showing that pre-treatment with NA, which removes all SA from the cell surface, completely blocked H-1PV cell binding/entry, whereas residual activity was observed in loss-of-function experiments targeting laminins.

Expression of laminin chains is tissue- and cell-specific. Laminins are a major component of the basement membranes, which are thin layers of the ECM that provide a substratum to which cells adhere and that are vital for tissue maintenance and survival. By modulating multiple signalling pathways, laminins influence fundamental cellular processes including adhesion, proliferation, migration, differentiation, and tumour metastasis[21,40,41].

The identification of *LAMC1* and therefore the laminin γ1 chain as a key determinant of H-1PV cell attachment and entry raised the question of what other laminin chains are involved in these events. Our competition experiments with soluble, purified laminins added to the cell culture medium indicate that different laminins containing the γ1 chain can efficiently decrease H-1PV infection. By contrast, laminin 332 (which contains the γ2 chain), collagen IV and fibronectin failed to arrest H-1PV infection. This is probably due to the different affinity that these molecules have for the virus capsid, which highlights the specificity of H-1PV–laminin interactions. This is also supported by our ELISA experiments which show that H-1PV can directly interact with laminins in the absence of any other factors. In agreement with the results obtained in cell culture experiments, H-1PV was found to interact preferentially with γ1-containing laminins rather than laminin 332, collagen IV or fibronectin. Results from the ELISA also show that pre-treatment of purified laminins with NA almost completely abolished the binding of H-1PV to laminins, which

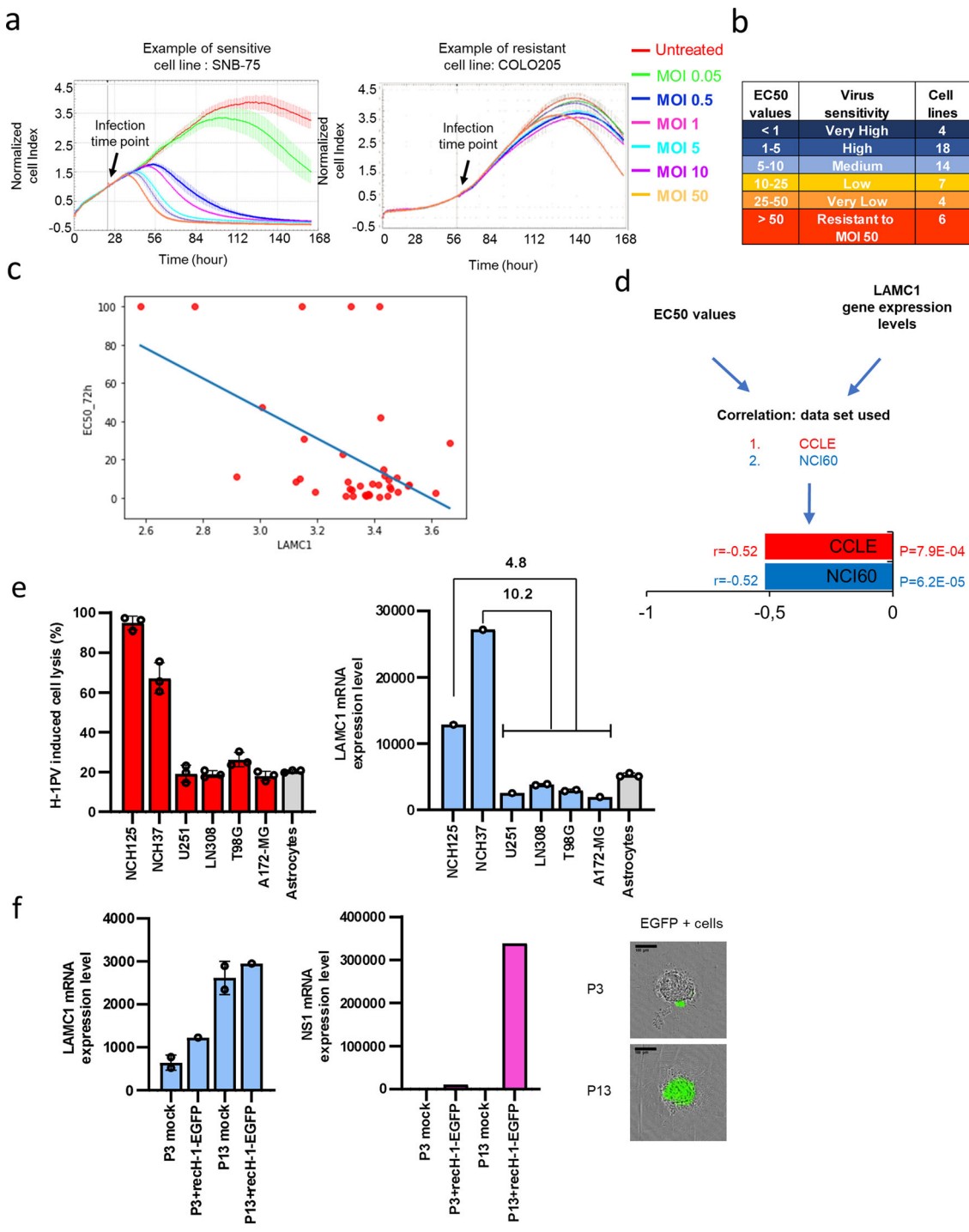

indicates that H-1PV-laminin binding occurs mainly via SA. However, our cell culture experiments do not exclude that other SA conjugated or not with other proteins may participate in the H-1PV cell attachment. Indeed, a residual H-1PV-binding activity at the cell surface was observed in *LAMC1* loss-of-function studies, as opposed with NA treatment that completely abolished this binding. The fact that the silencing of the *LAMC1* (or *LAMB1* gene) was less effective than NA treatment in reducing H-1PV infection may also be owing to the presence of a portion of laminin γ1 (or β 1) chain still remaining in these cells after gene silencing, as indicated by our Western blot analysis (e.g., as a consequence of incomplete gene silencing).

Taken together, these results provide evidence that sialylated laminins may serve as cell attachment receptor(s) of H-1PV for cell entry (Fig. 7). Binding of H-1PV to different laminins may explain the broad cellular tropism of H-1PV, which can infect a large number of cancer cells from different tissues.

Although laminins are not transmembrane receptor proteins, they may serve as docking points at the cell surface. Thus, it is possible that by acting as H-1PV attachment factors, laminins orchestrate interactions with one or more receptors on the cell surface, allowing H-1PV to cross the plasma membrane and penetrate the cell. The list of laminin cellular receptors includes various integrins (e.g., α1β1, α2β1, α3β1, α6β1, α6β4, α7β1, α9β1

**Fig. 6 Susceptibility of cancer cell lines to H-1PV oncolysis is directly correlated with *LAMC1* expression levels.** We tested 53 cancer cell lines from the NCI-60 panel for sensitivity to H-1PV infection using the xCELLigence Real-Time Cell Analyser (ACEA Biosciences, Inc). Cancer cell lines were left untreated or infected with the indicated H-1PV multiplicity of infection (MOI; expressed as plaque forming units (pfu) per cell). Cell growth was monitored in real time for a total of 168 h and expressed as normalised cell index (CI). **a** Kinetic response profiles of one H-1PV-sensitive (SNB-75) and one H-1PV-resistant (COLO205) cancer cell line are shown as representative examples; at least $n = 3$ biologically independent samples; error bar for the curve represents the mean values ± SD. **b** The CI values obtained by xCELLigence analysis were used to calculate the EC50 values (viral MOI that kills 50% of the cell population) at four time points after infection (24, 48, 72 and 96 h; see also Supplementary Fig. 12). Based on the EC50 value corresponding to 72 h, cancer cell lines were classified into the six indicated groups. **c** Scatterplot. Single points indicate the *LAMC1* expression levels in the 53 cancer cell lines that were screened for their susceptibility to H-1PV induced oncolysis (expressed as EC50). Source for the *LAMC1* expression levels is the NCI-60 COMPARE database. **d** *LAMC1* expression is robustly anticorrelated with EC50 value. Gene expression data from the NCI-60 (53 cell lines) and the CCLE (38 cell lines) panels were correlated with EC50 values (Pearson correlations). Bar plot shows the correlations, *r*, obtained in each dataset independently together with *P* values (both with significant correlations at $P < 1E-3$, null hypothesis: $r = 0$). These were generated in *R* (ggplot2 library). **e** *LAMC1* mRNA expression levels correlate with H-1PV efficacy of inducing cell lysis in glioblastoma cell lines. The indicated glioma-derived cell lines were infected with H-1PV (MOI 10 pfu/cell) for 72 h before being processed for an LDH assay to measure H-1PV-induced cell lysis (left panel). The independent experiment shown is repeated twice; $n = 3$ biologically independent samples. The total RNA from these six glioblastoma cell lines and astrocytes was isolated, and *LAMC1* mRNA expression levels were measured using NanoString analysis (right panel). Numbers on the top of the columns indicate gene expression fold changes between highly H-1PV-sensitive (NCH125 or NCH37) and poorly susceptible (U251, LN308, T98G, and A172-MG) cancer cell lines. The independent experiment is shown; $n = 1$ (NCH125, NCH37, U251 and A172-MG); $n = 2$ biologically independent samples (LN308 and T98G); $n = 3$ biologically independent samples (Astrocytes). **f** *LAMC1* mRNA expression levels correlate with both *NS1* mRNA expression levels and the transduction efficacy of H-1PV. The total RNA from untreated (mock) or recH-1PV-EGFP-infected P3 and P13 GBM 3D spheroids was isolated, and *LAMC1* (left panel) and NS1 (middle panel) mRNA expression levels were measured using NanoString analysis. A representative image of H-1PV transduction in P3 and P13 spheroids is shown in the right panel. For left panel the independent experiment is shown; $n = 2$ biologically independent samples for P3 and P13 mock treated whereas $n = 1$ for P3 and P13 infected with recH-1PV-EGFP. For middle panel, the independent experiment is shown; $n = 1$. For right panel, a single experiment with 10 replicates was carried out ($n = 10$). The transduction efficacy was analysed in these spheroids using the IncuCyte ZOOM™ live cell imaging system. The scale bar corresponds to 100 μm. Error bars for all data indicate the mean values ± SD. Source data are provided as Source Data file.

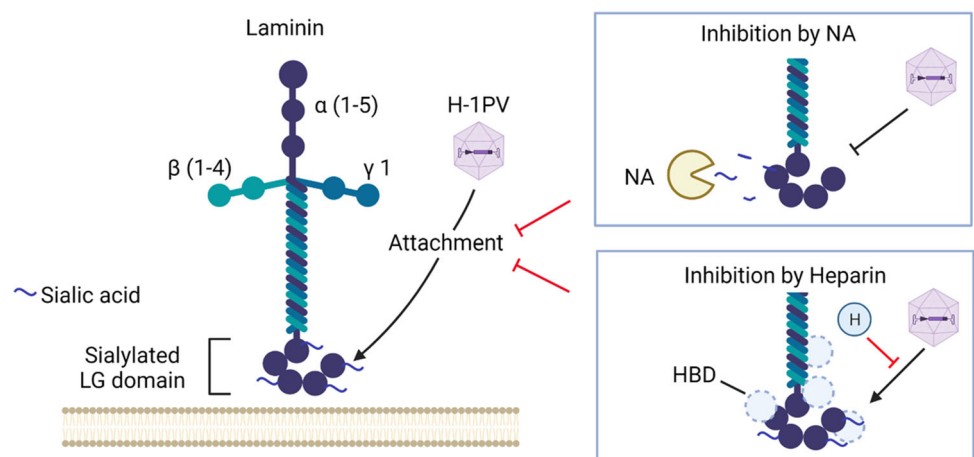

**Fig. 7 Visual abstract of the interaction between laminins and H-1PV.** Laminin γ1 chains form various heterotrimer complexes with α (1–5) chains and β (1–4) chains. In this study, we discovered that laminins are required for the attachment of H-1PV at the cell surface. H-1PV binding to laminins occurs via sialic acid moieties; neuraminidase (NA), which cleaves sialic acid, inhibits the interaction. Although heparin (H) is unable to bind directly to the H-1PV capsid (not illustrated), it prevents H-1PV from binding to laminins, probably by competing with the virus for the heparin-binding domains (HBD) present on laminins. The model was created by using Biorender.com.

and αvβ3), dystroglycan, galectins, HSPGs (perlecan and agrin), carbohydrate adduct of proteins-1 (HNK-1), Lutheran, syndecans, 67 kDa laminin receptor and sulphated glycolipids[21]. Different laminin isoforms bind specifically and with varying affinity to these molecules, forming distinct complexes. Given the tissue-specific distribution of laminins, we cannot exclude the possibility that different complexes may be required for H-1PV cellular uptake in different tumour types. It is also known that laminins interact with members of the galectin family[42]. Galectins are involved in the regulation of a broad range of host–microbial interactions and may facilitate virus infections[43]. Our siRNA library screening identified the *LGALS1* gene, encoding galectin-1, as a putative top activators of H-1PV life cycle. We postulate that galectins may participate in H-1PV cell attachment and entry in a complex with laminins at the cell surface. This hypothesis is supported by a previous study carried out in MVM showing that galectin-3 is required for efficient MVM uptake and infection[44]. Further studies are required to verify whether galectin-1 (and other members of the galectin family) in conjunction with laminins may represent another cell adhesion modulator of H-1PV.

The use of laminins as cell attachment factors is not exclusive for H-1PV alone. Other viruses interact with components of the ECM, and in particular with laminins, thereby recruiting other cell receptors to cross the plasma membrane. For instance, human papillomavirus internalisation into human keratinocytes is mediated by integrin α6β4 in cooperation with laminin 332. Similarly, laminins participate in vaccinia virus cell attachment. Human immunodeficiency virus type-1 interacts through gp120 and gp160 with the C-terminal heparin-binding site of fibronectin

as well as laminin and vitronectin. A broad spectrum of human pathogenic bacteria, such as *Haemophilus influenzae*, *Pseudomonas aeruginosa*, *Trypanosoma cruzi*, *Aspergillus fumigatus*, *Streptococcus agalactiae*, *Treponema pallidum*, *Leptospira interrogans* and *Talaromyces marneffei*, use laminins as cell adhesion factors (for a comprehensive review, see Singh et al.[21]). Binding of *H. influenzae* to laminins can be efficiently blocked by heparin, whereas the binding of *T. marneffei* to laminin is SA-dependent. Laminins are also an attractive target for a number of fungal and protozoal pathogens that use these ECM proteins as a docking site for cell attachment[21]. Interaction with laminins and other ECM constituents thus seems to be a common mechanism conserved throughout the evolution of different classes of microbes.

Heparin belongs to a family of complex carbohydrates known as the glycosaminoglycans. Owing to its highly negative charge, heparin can bind a wide array of positively charged biological materials. It is widely used as an anticoagulant in a range of clinical indications in which there is an increased risk of blood clot formation, e.g., during and after surgery, and in the treatment and prevention of deep vein thrombosis, pulmonary embolism and arterial thromboembolism. Heparin is also a known ligand for laminins and is required for laminin polymerisation and cell adhesion. Our competition experiments using increasing concentrations of soluble heparin showed that heparin inhibits H-1PV infection in a dose-dependent manner. It is therefore possible that H-1PV binding to laminin occurs through the heparin domains and that, in the presence of soluble heparin, steric hindrance prevents the binding of the virus to laminin (Fig. 7). Several heparin-binding sites have been mapped within laminin[35,36]. Heparin binds the laminin-type globular (LG) domains at the C-termini of the α chains with high affinity. This region mediates laminin attachment to the cell surface by binding to an array of molecules, including dystroglycan, syndecans, galactosylsulfatides and a number of integrins. It is worth noting that in present study, treatment with a peptide resembling the heparin-binding domain (HBD) within the LG domain of the laminin α chain decreased H-1PV transduction, which suggests that this domain is involved in the interaction between H-1PV and laminins. Moreover, ELISA recapitulated what was observed in cell culture experiments, showing that pre-treatment with heparin significantly decreased the H-1PV-binding affinity to laminins. As heparin was unable to bind to the virus capsid, these results suggest that heparin competes with H-1PV in binding to laminins and that the heparin-binding sites present in the laminins are presumably involved in the H-1PV–laminin interactions. However, the presence of SA moieties within heparin-binding sites remains to be demonstrated. Heparin is a highly charged molecule known to non-specifically bind to a number of molecules through a purely ionic interaction. It is possible that heparin by binding to laminins on different sites may also influence the conformation of laminins thus preventing H-1PV to access to SA moieties present in these molecules and thus hindering virus attachment at the cell surface. The discovery that H-1PV uses sialylated laminins as an attachment receptor and that heparin treatment impairs virus infectivity may have clinical consequences. Use of heparin as an anticoagulant during H-1PV treatment might reduce accessibility of the virus to cancer cells, thereby decreasing its efficacy. To this end, it is advisable to use an available alternative to heparin during H-1PV-based anticancer therapy.

Heparan sulphate proteoglycans (HSPGs) are ubiquitously expressed on the cell surface of all mammalian cells and interact with numerous effector molecules, including ECM proteins, growth factors and their receptors and cell–cell adhesion molecules[45]. Numerous viruses use HSPGs as cell attachment factors, including herpes simplex virus, HIV, vaccinia virus, Sindbis virus, Hepatitis C virus, dengue virus and adeno-associated virus[45–47]. We show that heparinase III treatment, which digests HSPGs, did not prevent H-1PV cell binding or entry, which argues against a role for HSPGs as additional cell attachment factors for H-1PV.

Our discovery that *LAMC1* is a key determinant of H-1PV infection raises the possibility of using *LAMC1* and possibly other laminin chains as biomarkers to predict the outcome of H-1PV-based therapies. We provide important evidence supporting this idea by proving a direct correlation between H-1PV oncolysis and *LAMC1* expression levels. We also found that the expression levels of other laminin members, such as *LAMA1*, *LAMA4* and *LAMB2*, were directly correlated to H-1PV oncolysis but to a lesser degree than that observed for *LAMC1*. Nevertheless, these results highlight the relevance of laminins for H-1PV oncolysis.

Interactions between cancer cells and laminins have a critical role at several steps of the complex cancer development process[48,49]. These interactions evoke signalling events inside the cells via interaction with the cellular receptors, which affect survival, differentiation, polarisation and migration. Other hallmark processes of cancer, such as angiogenesis, tumour invasion and metastasis, are accelerated. Laminins have already been proposed as prognostic markers of various cancers[48]. Our analysis shows that upregulation of *LAMC1* and *LAMB1* gene expression is a marker of poor prognosis for certain tumours. Various other studies suggest that *LAMC1* upregulation in certain types of tumour, such as hepatocellular carcinoma[50], uterine carcinoma[51] and GBM[52], is associated with poor prognosis and tumour progression. Our protein tissue microarray performed on GBM biopsies ($n = 110$) showed a varied distribution of laminin γ1 chains across the samples. We speculate that tumours expressing high levels of *LAMC1* may be more susceptible to H-1PV infection. Conversely, cancers expressing low levels of laminins will most likely respond poorly to H-1PV treatment simply because they are less accessible to viral infection. On the other hand, normal tissues also express laminins although often at lower level than their tumour counterparts. This would explain the ability of H-1PV to also infect normal cells. These infections however, do not result in the lysis of the cells or in the production of new progeny particles (abortive infections), because normal cells lack important determinants needed for H-1PV replication[53].

The identification of biomarkers that could predict the outcome of a certain anticancer treatment is certainly an important area of investigation. The oncolytic virus life cycle is strictly dependent on many cellular factors. We showed that laminins are involved in H-1PV cell attachment and entry but other cellular factors acting at the entry and post-entry levels are also required for H-1PV productive infection. Although important for H-1PV infection, the only analysis of the laminins in tumours may be not sufficient to predict the outcome of H-1PV treatment. In our opinion, only the analysis of a large number of H-1PV modulators (positive or negative) on tumour specimens can help the identification of those patients most likely responding to H-1PV-based therapies. Many of these modulators remain to be identified.

In conclusion, our study reveals laminins as an important new player in the H-1PV life cycle as cell attachment factors. The discovery of laminins as a host cell factor involved in H-1PV attachment may facilitate the identification of other modulators involved in H-1PV cell entry (e.g., proteins interacting with laminins on the cell surface). This will not only improve our knowledge of H-1PV biology but also pave the way for the identification of genes that could serve as predictive and prognostic markers to inform clinical management and improve patient survival.

## Methods

**Cell cultures**. The cervical carcinoma-derived HeLa cell line was a gift from Angel Alonso (German Cancer Research Center, Heidelberg, Germany). The transformed human embryonic kidney HEK293T cell line was obtained from the American Type Culture Collection. The low-passage-number GBM-derived NCH125 and NCH37 cell lines were provided by Karsten Geletneky (Heidelberg University Hospital, Heidelberg, Germany). The pancreatic ductal adenocarcinoma (PDAC)-derived BxPC3 cell was procured from Tumorbank (German Cancer Research Center). The colon colorectal carcinoma-derived HCT116, the lung adenocarcinoma-derived A549 and GBM-derived U251cell lines, were obtained from the National Cancer Institute (NCI; Rockville, MD, USA). GBM-derived LN308, T98G and A172-MG were obtained from Iris Augustin's laboratory (DKFZ, Heidelberg, Germany). Normal astrocytes isolated from human brain (cerebral cortex) were purchased from ScienCell Research laboratories (Carlsbad, CA.

HeLa, HEK293T, NCH125, NCH37, U251, LN308, T98G, A172-MG and LAMC1-KD (the engineered HeLa cell line established in this study, in which *LAMC1* expression was down-regulated by CRISPR-Cas9 technology) were grown in Dulbecco's modified Eagle's medium (DMEM) supplemented with 10% foetal bovine serum (FBS; Gibco, Thermo Fisher Scientific, Darmstadt, Germany) and 2 mM L-glutamine (Gibco). For LAMC1-KD, 2 μg/ml puromycin was added to the medium for clone selection, expansion and maintenance. The HCT116, A549 and BxPC3 cancer lines were grown in RPMI medium with the addition of 10% FBS and 2 mM L-glutamine. The 53 cancer lines belonging to the NCI-60 panel (Supplementary Table 2) used in this study were all grown in RPMI medium with the addition of 10% FBS and 2 mM L-glutamine. The human astrocytes were grown in astrocyte medium containing 1% astrocyte growth supplements (ScienCell), 2% FBS and 1% penicillin/streptomycin solution. All cell lines were grown at 37°C in 5% $CO_2$ and 90% humidity. The identity of the cell lines was verified by a human cell authentication test (Multiplexion GmbH, Mannheim, Germany), and the lines were frequently tested and confirmed to be free of mycoplasma contamination using a VenorGEM OneStep Mycoplasma contamination kit (Minerva Biolabs, Berlin, Germany).

Patient-derived orthotropic xenograft models (P3, P13) were derived from human GBMs obtained from the Department of Neurosurgery, Haukeland University Hospital in Bergen Norway. Patients provided informed consent and tumour collection was approved by the Regional Ethical Board at the Haukeland University Hospital in Bergen. Tumours were primary IDH1wt GBMs based on neuropathological diagnosis and genomic analysis. GBM tumour tissue was serially transplanted in NOD/SCID mice as previously described[54]. P3 and P13 spheroids of uniform size were derived from purified primary single tumour cells seeded in low attachment 3D 384-well plates (primeSurface 384U plate, SBio, Japan) at a density of 1000 cells/well in 40 μl of spheroid culture medium consisting of DMEM medium supplemented with 10% FBS, 2 mM L-Glutamine, 0.4 mM non-essential amino-acid solution and 100 U ml$^{-1}$ Pen-Strep (Invitrogen, USA). 3D spheroids were reformed under orbital agitation at 37°C under 5% $CO_2$ atmospheric oxygen.

**High-content siRNA library screening of H-1PV transduction**. The human druggable genome small interfering RNA (siRNA) Set 4.0 library comprising siRNA pools (4 siRNAs/pool, 1 pool/gene) targeting 6961 cellular genes was purchased from Qiagen (Hilden, Germany). The library was reverse transfected in HeLa cells grown in Greiner μClear 96-well microplates using the INTERFERin reagent (Polyplus-transfection® SA, Illkirch, France). The high efficiency of transfection in this cell line makes it amenable to this kind of study. The high-throughput transfection protocol was optimised to reach 90–95% transfection efficiency with minimal toxicity. The screening was performed in technical triplicate. The same cell passage ($n = 3$ after thawing), serum and transfection agent batch were used for all plates to limit biological variability. The following internal control siRNAs were used in each microplate to control inter-plate and day-to-day variability: (i) NS1-siRNA5: 5'GAATGGGTTACCAATCTACC3', a siRNA targeting the NS1 coding region, used as a positive control as the NS1 protein is essential for virus transduction; (ii) scramble siRNA: 5'AATTCTCCGAACGTGTCACGT3', a non-targeting siRNA, used as a negative control (Qiagen); and (iii) *polo-like kinase-1* gene (*PLK1*) siRNA: 5'CAACCAAAGTCGAATATGA3', targeting the *PLK1* gene, used as transfection efficiency control because the silencing of this gene leads to cell death. Two sets of cells were reverse transfected with control siRNAs or the druggable siRNA library comprising siRNA pools individually targeting a total of 6961 different genes and then grown for 48 h to allow efficient gene silencing. One set of cells was left untreated to control the intrinsic cytotoxicity of every transfected siRNA pool, whereas the other set of cells was infected with recH-1PV-EGFP used at multiplicity of infection (MOI) of 0.35 pfu/cell This replication-deficient recombinant parvovirus shares the same capsid as the wild-type virus, but harbours the *EGFP* reporter gene under the control of the natural parvoviral P38 promoter, which replaces part of the *VP* gene unit encoding for the VP1 and VP2 capsid protein[55]. This promoter is specifically activated by the parvovirus NS1 protein, and therefore its expression, which correlates with the EGFP signal, is a direct measure of the transduction abilities of H-1PV. At 24 h post infection, plates were then fixed, stained with 4′,6-diamidino-2-phenylindole (DAPI) and subjected to fluorescence imaging to quantify H-1PV transduction efficiency by determining the percentage of EGFP-positive cells. A Freedom EVO liquid handling workstation (Tecan, Männedorf, Switzerland) was used for plate distribution of siRNAs,

transfection reagent, cells and virus. High-throughput cell imaging was carried out with the INCELL1000 HCS epifluorescent microscope (GE Healthcare Life Sciences, Freiburg, Germany), by analysing an average of 25,000 cells per microwell. The Multi Target Analysis module (IN Cell Investigator software, GE Healthcare Life Sciences, Chicago, IL, USA) was used to segment cell nuclei (DAPI staining) and quantify EGFP colocalization to determine the percentage of transduced cell. We applied this cellular segmentation protocol to determine the EGFP signal intensity per analysed DAPI-stained cell in the non-infected and infected cells. A EGFP-positive cell corresponds to a cell exhibiting a signal superior or equal to the 2% top signal intensities detected in the EGFP channel for the non-infected cells. Single-cell data were analysed with the RReportGenerator software, which also determined statistical significance[56]. The EGFP signal obtained in cells transfected with control scramble siRNA was used as a baseline to normalise the percentage of the EGFP signal obtained in cells individually transfected with siRNA pools for each of the 6961 target genes.

**Gene ontology enrichment analysis of top candidates**. The 151 genes (H-1PV putative activators) identified by the high-throughput druggable genome siRNA library screen were subjected to gene ontology enrichment analysis as described below:

A) Ingenuity® pathway analysis (IPA). The genes were uploaded to the IPA tool (Qiagen; https://www.qiagenbioinformatics.com/products/ingenuity-pathway-analysis)[57]. First, we carried out canonical pathway analysis, which localises genes to a canonical pathway based on previous knowledge of their association stored in the Ingenuity® Knowledge Base library of canonical pathways (Supplementary Figure 1a). The statistical significance of the association between a set of genes and a canonical pathway was measured by Fisher's exact test ($P < 0.05$). Next, we performed cellular component gene ontology (GO) analysis, which classified genes based on their subcellular location according to the GO Ingenuity® Knowledge Base (Supplementary Figure 1b).

B) PANTHER (Protein ANnotation THrough Evolutionary Relationship) analysis. The 151 genes identified as putative top activators by the siRNA library were submitted to the PANTHER classification system (http://www.pantherdb.org/)[58] to annotate them according to the cellular functions assigned by PANTHER Protein Class ontology terms (Supplementary Figure 1c). The software selected 99 PANTHER protein class hits after the analysis.

**Virus production**. Wild-type H-1PV was produced, purified and titrated as previously described[27]. Recombinant H-1PV (recH-1PV-EGFP) harbouring the *EGFP*-encoding gene was produced according to the protocol described in El-Andaloussi et al.[55,59]. Lentivirus expressing *LAMC1*-specific guide RNA (lenti-CRISPRv1-sgLAMC1) was produced as previously described[60]. In brief, HEK293T cells were transfected with lentiCRISPRv1-sgLAMC1, psPAX2 (Addgene, Cambridge, MA, USA) and pMD2.G (Addgene) plasmids at the ratio (2:1.5:1) using the Lipofectamine LTX and Plus reagent (Life Technologies Europe, Bleiswijk, Netherlands) according to the manufacturer's protocol. At 70 h post transfection, the culture medium containing lentiviral particles was removed and centrifuged, and cell debris was discarded. The supernatant was filtered through a 0.45 μM filter (Millipore Steriflip HV/PVDF; Merck Millipore, Burlington, MA, USA), concentrated 100× with PEG-it (BioCat Gmbh, Heidelberg, Germany) and re-suspended in DMEM medium containing 10% FBS, 2 mM L-glutamine and 1% bovine serum albumin. Lentivirus aliquots were stored at −80°C. Lentivirus titration was carried out using Global UltraRapid Lentiviral Titer Kit (SBI System Biosciences, Palo Alto, CA, USA).

**Plasmid constructions**. The lentiCRISPRv1 plasmid[60] was purchased from Addgene. The *LAMC1*-specific single guide RNA (sgRNA) **G**ATGGACGAGTG CACGGACGA was designed using the online CRISPR design tool (http://crispr.genome-engineering.org/) and then cloned into a BsmBI digested lentiCRISPRv1 plasmid. A G nucleotide (in bold) was added to the sgRNA for efficient recognition by the U6 promoter. The resulting recombinant lentivirus expressing *LAMC1*-specific guide RNA was designated as lentiCRISPRv1-sgLAMC1. For gain-of-function experiments, the pCAG-LAMC1-S/MAR expression vector was generated. The full-length *LAMC1* gene encoding laminin γ1 cloned into pTriEX-1 (a generous gift of Winfried Stocker, EuroImmun, Lubeck, Germany) was used as a template for the PCR. PCR amplification was performed using CloneAmp HiFi PCR Premix (Takara Bio, Mountain View, CA, USA) using the primers listed in Supplementary Table 3. The PCR fragment was then cloned into *Bgl*II-digested pCAG-S/MAR (kindly provided by Richard Harbottle, German Cancer Research Center) using the In-Fusion HD cloning kit (Takara Bio-Europe, Saint-Germain-en-Laye, France).

**Generation of LAMC1-KD cell line**. For the generation of LAMC1-KD cells, 5 × 10⁴ HeLa cells were seeded in a 24-well plate and infected with 9.5 × 10⁶ IFU (100 μl) of lentivirus particles harbouring the lentiCRISPRv1-sgLAMC1 DNA. After 24 h, infection was repeated with the same amount of lentivirus and cells grown for an additional 24 h. DMEM medium supplemented with 10% FBS, 2 mM

L-glutamine, 1% BSA and 2 μg/ml of puromycin was added for the selection of infected cells. Single-cell clones were expanded and confirmed for *LAMC1* silencing by western blot analysis.

**Protein extraction and analysis**. Cells were harvested from dishes by gently scraping with a rubber policeman directly in the culture medium. Cells were collected by centrifugation and washed with ice-cold PBS. After centrifugation, cell pellets were suspended in five volumes of lysis buffer (50 mM Tris, pH 8, 200 mM NaCl, 0.5% NP-40, 1 mM dithiothreitol) containing protease inhibitors (complete EDTA free; Roche, Mannheim, Germany) and lysed on ice for 30 min. Cellular debris was removed by centrifugation ($12,000 \times g$ for 15 min at 4 °C), and protein concentration in cell lysates was measured by bicinchoninic acid assay (Thermo Fisher Scientific), according to manufacturer's instructions. Sodium dodecyl sulfate polyacrylamide gel electrophoresis (SDS-PAGE) analysis was performed on 25–50 μg of total protein extract. After separation, proteins were transferred to a Hybond-P membrane (GE Healthcare, Freiburg, Germany). Immunoblotting was carried out with the following antibodies: mouse monoclonal anti-β-tubulin (clone TUB 2.1; Sigma-Aldrich, St. Louis, MO, USA) used at 1:2000 dilution; mouse monoclonal anti-laminin γ1 (clone B-4; Santa Cruz Biotechnology, Heidelberg, Germany) at 1:500 dilution; mouse monoclonal anti-laminin β1 (clone D-9; Santa Cruz Biotechnology), at 1:500 dilution. After incubation with horseradish peroxidase-conjugated secondary antibodies (Santa Cruz Biotechnology), the membrane was incubated with Western Blot Chemiluminescence Reagent *Plus* (Perkin Elmer Life Sciences, Waltham, MA, USA) and exposed to Hyperfilm™ ECL radiographic films (GE Healthcare, Buckinghamshire, UK).

**Virus capsid labelling**. Iodixanol-purified full H-1PV capsids ($4.4 \times 10^9$ pfu/ml) were conjugated with Alexa Fluor™ 488 using Alexa Fluor™ 488 Microscale Protein Labeling Kit (Thermo Fisher Scientific) according to the supplier's manual. In brief, 100 μl of solution containing the virus was mixed with 10 μl of 0.1 M bicarbonate buffer (pH 8.3) and 10 μl of Alexa Fluor™ 488 dye re-suspended in water. After 1 h incubation at 22 °C, the labelling reaction was stopped by adding 150 mM (final) hydroxylamine. The labelled virus was recovered using a Zeba™ Spin Desalting Column, 7 K MWCO, 0.5 ml (Thermo Fisher Scientific).

**Colocalization of labelled virus with laminins**. NCH125 cells were seeded at a density of $3.5 \times 10^3$ cells/spot on spot slides and grown in 50 μl of complete cellular medium. After 24 h, cells were placed on ice for 15 min and then infected with 1 μl of labelled H-1PV used at MOI 50 (pfu/cell) in a total of 70 μl complete cellular medium. At 2 h post infection, the cells were fixed with 2% paraformaldehyde on ice for 5 min and then stained with pan-laminin antibody (Cat. no. L9393, Sigma-Aldrich Chemie GmbH, Steinheim, Germany) at dilution 1:25 for 1 h. Images in green channel (labelled virus) and red channel (laminins) were acquired with a confocal microscope (Leica TCS SP5 II, Wetzlar, Germany). Analysis of colocalization between labelled virus and laminins was carried out using the ImageJ plugin RGB profiler (https://imagej.nih.gov/ij/plugins/rgb-profiler.html) created by Christophe Laummonerie and Jerome Mutterer. The pixel size was set to 34 nm, the objective used for the analysis was ×63/1.4 NA (numerical aperture), and the imaging was performed at the resolution limit of 200 nm.

**H-1PV cell uptake**. H-1PV cell uptake was determined by analysing the amount of viral genome associated to the cells. Cells were washed two times with PBS in order to remove unbound H-1PV particles before to be processed for viral DNA extraction and quantification as described below.

**H-1PV DNA extraction and quantification**. Cells were subjected to three snap freeze–thaw cycles to release cell-associated viral particles. Viral DNA was purified both from an aliquot of the virus used as inoculum and from cell lysates using the QiAamp MinElute Virus Spin kit (Qiagen) according to the manufacturer's instructions. Viral DNA was quantified using a parvovirus-specific qPCR as previously described[55]. Primers and probe used for the qPCR are described in Supplementary Table 3.

**siRNA-mediated knockdown experiments for H-1PV cell uptake analysis**. HeLa, HCT116 and A549 cell lines were seeded at a density of $4 \times 10^4$ cells/well in a 24-well plate and grown in 500 μl of complete cellular medium without antibiotics. After 24 h, cells were transfected with siRNAs (5–10 nM) in a serum-free medium using Lipofectamine RNAimax (Thermo Fisher Scientific) according to the manufacturer's instructions. The following siRNAs were used: LAMC1_1 (SI00035742), LAMC1_5 (SI02757475), LAMB1_4 (SI00035707) and LAMB1_9 (SI05109174), with AllStars Negative siRNA (SI03650318) used as negative control (all purchased from Qiagen). After 24 h, the medium was changed, and cells were grown for an additional 24 h to allow efficient gene silencing. The culture medium was then removed and replaced with 0.2 ml serum-free medium containing H-1PV at MOI 1 pfu/cell. Infection was performed for 4 h at 37 °C to allow cell surface binding and internalisation of viral particles. H-1PV cell uptake was performed as described above.

**Drug/enzyme pre-treatment for H-1PV cell uptake analysis**. Heparin (Cat. No. H4784), NA (Cat. No. N2876) and heparinase III (Cat. No. H8891) were purchased from Sigma-Aldrich Chemie GmbH. Heparin stock solution was freshly prepared and used on the same day. NA and heparinase III stock solutions were prepared, aliquoted and stored at −20 °C before use. HeLa, BxPC3 and NCH125 cells were seeded at a density of $4 \times 10^4$ cells/well in 24-well plates and then pre-treated with increasing amounts (10, 25, 50, 100 μg/ml) of heparin or NA (0.1 U/ml) or heparinase III (0.1 U/ml) for 24 h. Before being used for infection, H-1PV (MOI 1 pfu/cell) was preincubated with the same concentrations of heparin, NA or heparinase III for 15 min at room temperature. The cell binding/entry assay was performed for 4 h at 37 °C. H-1PV cell uptake was determined as described above.

**Antibody competition assay**. For the antibody competition assay, HeLa cells, seeded at a density of $2 \times 10^4$ cells/well in a 24-well plate, were grown in 500 μl of complete cellular medium without antibiotics. After 24 h, cells were incubated for 45 min on ice with 10 μg/ml of mouse monoclonal control anti-IgG (isotype control) (Millipore, Temecula, CA, USA) or mouse monoclonal anti-laminin γ1 (clone B-4; Santa Cruz Biotechnology) in DMEM medium supplemented with 10% FBS. At the end of incubation, the medium was removed and replaced with 0.1 ml serum-free medium containing H-1PV (MOI 0.25 pfu/cell). The virus binding assay was first performed for 30 min on ice. Cells were then washed twice with PBS to remove unbound virus particles and further incubated in fresh complete cellular medium for 1 h at 37 °C to allow internalisation of virus particles. Cells were then processed for H-1PV cell uptake quantification.

**Gain-of-function experiments for H-1PV cell uptake analysis**. HeLa and LAMC1-KD cell lines were seeded in 24-well plates and transiently transfected either with plasmid alone or plasmid carrying *LAMC1* in serum-free medium using lipofectamine LTX (Thermo Fisher Scientific) according to the manufacturer's instructions. At 4 h post transfection, the medium was replaced by complete cellular medium and cells were grown for an additional 44 h. Virus cell uptake was carried out by infecting the cells with H-1PV (MOI 100 pfu/cell) for 4 h at 37 °C. H-1PV cell uptake was determined as described above.

**Virus cell surface binding assay**. After siRNA transfection, the culture medium was removed and replaced with 0.2 ml serum-free medium containing H-1PV at MOI 1 pfu/cell. Infection was performed for 2 h at 4 °C to allow only cell-surface virus binding. H-1PV cell uptake was quantified as before.

**H-1PV cell transduction assay**. Cells were washed once with PBS, fixed in 3.7% formaldehyde for 5 min, permeabilized with 1% Triton X-100 for 10 min and stained with DAPI for 2 min. Fluorescence images of EGFP-positive cells were acquired with a BZ-9000 fluorescence microscope (Keyence Corporation, Osaka, Japan) with either ×10 or ×20 objective, as described in figure legends. DAPI staining was used to visualise the number of nuclei (cells). The percentage of EGFP-positive cells was calculated by counting at least 1500 cells.

**siRNA-mediated gene knockdown experiments for H-1PV transduction assay**. HeLa cells were seeded at a density of $2 \times 10^4$ cells/well in a 24-well plate and grown in 500 μl of complete cellular medium without antibiotics. For laminins, we used the same siRNAs described for the virus cell uptake assay. SiRNA transfection was performed as described above. At 46 h post transfection, cells were infected with recH-1PV-EGFP (TU 1, EGFP/cell) and grown for an additional 24 h. H-1PV cell transduction activity was determined as described above.

**Pre-treatment with heparin for H-1PV transduction assay**. HeLa, BxPC3 and NCH125 cells were seeded at a density of $4 \times 10^4$ cells/well in 24-well plates. After pre-treatment with heparin (100 μg/ml) for 18 h, cells were infected with recH-1PV-EGFP (TU 0.5, EGFP/cell) and grown for additional 24 h. H-1PV cell transduction was determined as described above.

**Treatment with peptide derived from the HBD of laminin α 4**. The HBD in the LG domain of laminin α chain, spanning amino acids 1408–1434 and with the sequence PLFLLHKKGKNLSKPKASQNKKGGKSK[61], was synthesised by BioCat GmbH. The peptide was reconstituted in 100% dimethyl sulfoxide at a final concentration of 50 mM. HeLa cells were seeded at a density of $4 \times 10^3$ cells/well in 96-well plates. The next day, cells were preincubated with laminin α four derived peptide (200 μM) in 100 μl of DMEM medium containing 10% FBS for 1 h before infecting cells with recH-1PV-EGFP (TU 0.5, EGFP/cell) for 24 h. The cells were then processed as described above to determine H-1PV cell transduction activity.

**Treatment with soluble laminins, fibronectin or collagen**. NA and collagen IV were purchased from Sigma-Aldrich Chemie GmbH, and fibronectin from Merck KGaA (Darmstadt, Germany). All laminins (LN111, LN121, LN211, LN221, LN411, LN421, LN511, LN521 and LN332) were obtained from BioLamina AB (Stockholm, Sweden). HeLa cells were seeded at a density of $2 \times 10^3$ cells/well in 96-well plates. After 24 h, cells were pre-treated with NA (0.2 U/ml), fibronectin

(2 µg/ml), collagen IV (2 µg/ml) or laminins (2 µg/ml) in 100 µl of DMEM medium containing 10% FBS. At 24 h post treatment, cells were infected with recH-1PV-EGFP (TU 0.5, GFP/cell) and grown for an additional 27 h. Cells were then washed once with PBS, fixed in 3.7% formaldehyde for 5 min, permeabilized with 1% Triton X-100 for 10 min and stained with DAPI for 2 min. Fluorescence images of EGFP-positive cells were acquired with a BZ-9000 fluorescence microscope from Keyence Corporation with ×10 objective.

**Gain-of-function experiments for H-1PV transduction assay.** We seeded $2 \times 10^4$ HeLa cells in a 24-well plate and then transfected them with either vector alone or vector expressing *LAMC1*. At 46 h post transfection, cells were infected with recH-1PV-EGFP (0.25 TU, EGFP/cell) and grown for an additional 24 h before being processed as described above.

**H-1PV-binding assay with solid-phase ELISA.** Microtiter plates (96-well plates, Nunc-Immuno MaxiSorp surface plate; Thermo Fisher Scientific) were pre-coated overnight at 4°C with 1 µg/well of purified LN211, LN221, LN441, LN421, LN332 laminins (BioLamina) diluted in a mixture of Dulbecco's phosphate-buffered saline supplemented with calcium and magnesium salts and 50 mM sodium carbonate (pH 9.6) (1:1 ratio, total volume 50 µl). As controls, wells were also pre-coated with 2.56 µg/well of collagen type IV (Sigma), human plasma fibronectin (Merck Millipore), or albumin bovine fraction V (BSA; SERVA) diluted in 50 mM sodium carbonate (pH 9.6). For the digestion of SA from laminins, NA (0.1 U/ml) was added to the well. For competition experiments, heparin (Cloud-Clone Corp.) at a concentration of 20 µg/ml was added. As a control, wells were also pre-coated with 20 µg/well of BSA-conjugated-heparin. Washing was performed four times in between all the steps with PBS 0.1 % Tween 20. Blocking was carried out for 2 h at room temperature with 2% BSA diluted in PBS. H-1PV particles ($7 \times 10^8$ pfu/well) diluted in 2% BSA containing PBS were added to the wells and incubated overnight at 4°C. Bound virus particles were detected at room temperature using a mouse anti-H-1PV capsid antibody[62] used at the dilution of 1:500 for 2 h, followed by peroxidase-conjugated goat anti-mouse antibody (1:500; Santa Cruz Biotechnology) for 2 h, and 3,3′,5,5′-tetramethylbenzidine (TMB) substrate solution (Pierce™, Thermo Fisher Scientific) incubated for 5 min. The reaction was stopped using the stop solution for TMB substrate (N600; Thermo Fisher Scientific). Final readings were carried out at 450 nm using the Thermo Multiskan EX Microplate photometer (Thermo Fisher).

**Cell viability assays.** In siRNA knockdown experiments, the siRNAs described above were spotted onto 96-well plates (one siRNA per well) in triplicate and then reverse transfected in $2.5 \times 10^3$ HeLa cells. After 24 h, the cell culture medium was exchanged, and cells were grown for an additional 48 h before being infected with H-1PV (MOI 0.25 pfu/cell). At 72 h post infection, cell viability was measured using the CellTiter-Glo 2.0 assay (Promega, Madison, WI, USA). In brief, cells were brought to room temperature, and the cell culture medium was replaced with 100 µl of CellTiter-Glo 2.0 reagent (diluted 1:2 in cell culture medium). Plates were placed on an orbital shaker for 2 min and then incubated with the solution for 10 min. Luminescence was read on a LB 943 Mithras[2] plate multimode reader (Berthold Technologies, Bad Wildbad, Germany). For the heparin treatment experiment, HeLa cells were seeded into a 96-well plate at a density of $2.5 \times 10^3$ cells per well. After 24 h, heparin (100 µg/ml) was added to the medium and cells were grown for an additional 24 h before being infected with H-1PV (MOI 1 pfu/cell) that had previously been preincubated with heparin for 15 min at room temperature. Cell viability was measured 96 h post infection using CellTiter-Glo 2.0, as described above.

**Lactate dehydrogenase assay.** Cancer cell lines were plated in 96-well plates at a density of 4000 cells/well and grown for 24 h. Cells were then infected with H-1PV and grown for an additional 72 h in DMEM medium supplemented with 5% of heat-inactivated bovine serum (100 µl/well). Virus-induced cell lysis was determined by the amount of lactate dehydrogenase released into the culture medium using the Cytotox 96® non-radioactive cytotoxicity assay kit (Promega, Mannheim, Germany), as previously described[19].

**NCI-60 cancer cell line screening: cell proliferation assay.** Cell proliferation was monitored in real time using the RTCA-MP xCELLigence system (ACEA Biosciences Inc., San Diego, CA, USA) including $6 \times 96$-well electronic microtiter plate modules. Cells were seeded on a 96-well E-Plate at a density of 4000–16000 cells/well (according to cell doubling rate). Cells were infected 24–72 h later with increasing amounts of wild-type H-1PV, ranging from MOI 0.05 pfu/cell to 50 pfu/cell (0, 0.05, 0.25, 0.5, 1, 5, 10 and 50). Growth of untreated and H-1PV-infected cells was monitored in real time for 5–7 days every 30 min and expressed as normalised cell index (CI), a parameter proportional to the number of attached cells per well and, therefore, strictly correlated with cell proliferation rate. The experiment was carried out at least three times for every condition. Six leukaemia cancer cell lines that grow in suspension were excluded from the screening, because of incompatibility with the xCELLigence system, which monitors the growth of only adherent-growing cell lines.

**RNA isolation.** GBM-derived cell lines were grown in six-well plates in DMEM medium supplemented with 10% FBS and 2 mM L-glutamine at a density of 500,000 cells/well. After 20 h, cells were collected. P3 and P13 GBM primary 3D organotypic spheroid cultures were grown according to culture conditions mentioned in cell cultures section. After 48 h, spheroids were mock treated or infected with recH-1PV-EGFP (TU 100, EGFP/cell). Experiments for each condition was performed in ten replicates and cells were pooled. Total RNA was isolated from cell pellets and spheroid pellets using the Quick-RNA miniprep kit (Zymo Research Irvine, CA, USA) according to the manufacturer's instructions.

**nCounter gene expression analysis.** nCounter target gene expression analysis was carried out as recommended by NanoString Technologies (Seattle, WA, USA). The nCounter technology allows for multiplexed gene expression analysis based on simultaneous hybridisation and digital quantification of fluorescently labelled oligonucleotide probes[63]. The probe set used for the analysis is described in Supplementary Table 4.

All RNA samples were quantified using a Qubit™ RNA HS assay kit (Thermo Fisher Scientific, USA) and quality control was performed using the Agilent RNA6000 Nanokit on an Agilent 2100 Bioanalyzer system (Agilent, Santa Clara, CA, USA). Samples were subjected to overnight hybridisation. In brief, 50 ng of total RNA were used as input material for probe set hybridisation at 65 °C. Up to 7 µl of total RNA samples were combined with 2 µl of nCounter customised TagSet, 5 µl of hybridisation buffer and 0.5 µl Probe A plus 0.5 µl Probe B for a total reaction volume of 15 µl. Samples were incubated for 20 h, cooled down to 4 °C and then purified and immobilised on a cartridge. Readout of the experiment was performed using the SPRINT™ Profiler from NanoString Technologies. Normalisation and evaluation of data were carried out using the nSolver Analysis Software (version 4.0) provided by NanoString Technologies (https://www.nanostring.com/products/analysis-software/nsolver). Stably expressed reference genes were chosen for normalisation based on the Normfinder method[64].

**IncuCyte ZOOM™ assay.** At 48 h post seeding, spheroids were infected with recH-1PV-EGFP (TU 100, EGFP/cell) in 40 µl of spheroid culture medium without FBS. The kinetics of virus infection were measured by acquiring images of spheroids at intervals of 4 h for a total of 120 h at ×10 magnification with IncuCyte ZOOM™ live cell imaging system (Essen BioScience, Ann Arbor, MI, USA). Experiments were performed ten times.

**Evaluation of laminin γ1 expression in normal tissues, primary and recurrent GBM biopsies.** GBM tissue microarrays were prepared from paraffin blocks of GBM biopsies obtained at Haukeland University Hospital, Bergen, Norway (project approved by the regional ethical committee). The GBM tissue microarray comprises 110 different GBM patient biopsies obtained from 61 primary and 49 recurrent GBMs. Furthermore the array also contains 12 biopsies from normal tissues, namely brain, liver and tonsil (four for each organ). Immunohistochemical staining was performed according to the protocol described previously[65] using laminin γ1 antibody (sc-17751, Santa Cruz) at a dilution of 1:200 and biotinylated anti-mouse antibody (Vector Laboratories) at a dilution of 1:100. Numbers of positive cells were quantified digitally as described previously[66].

**Determination of the EC50 value relative to H-1PV infection in the NCI-60 cancer cell line screening.** A two-step data analysis approach[67] was applied to derive, for each NCI-60 cell line, EC50 values (virus concentration killing 50% of cells) at 24, 48, 72 and 96 h post infection using the normalised CI values obtained with the xCELLigence system as an input. In step one of this approach, one-way analysis of variance (ANOVA) followed by post hoc Dunnett contrast testing of the contrast 'MOI 0 versus MOI 50' was carried out to assess whether H-1PV amount has a consistent effect on normalised CI. We concluded that there was no consistent effect if (a) ANOVA failed to demonstrate a global effect of H-1PV amount on normalised CI ($p > 0.05$) or (b) a global but inconsistent effect was revealed by ANOVA ($p \leq 0.05$) followed by post hoc Dunnett contrast testing ($p > 0.05$). No Dunnett contrast testing was needed and was thus not performed for case (a). If no consistent effect was found in step one, no EC50 value was reported for the respective combination of NCI-60 cancer cell line and post infection time point.

In step two of this approach, the EC50 value was computed by fitting the four-parameter log-logistic model to the concentration-response data (concentration: H-1PV amount; response: normalised CI) that was obtained for the respective combination of NCI-60 cancer cell line and post infection time point. As negative values for the normalised CI are not biologically meaningful, the lower asymptote of the four-parameter log-logistic model function was restricted to be ≥ 0. No EC50 value was reported in four situations: (1) if the estimated EC50 value exceeded the maximum H-1PV amount tested (e.g. MOI 50 pfu/cell), it was considered unreliable and was thus not reported; (2) if the distance between the lower asymptote c and the upper asymptote d of the fitted concentration-response curve was too small (e.g., if $c > 0.7*d$), the observed effect of H-1PV amount on normalised CI was considered to be irrelevant and the EC50 estimate was thus not reported; (3) if an increasing concentration-response curve, which is not interpretable from a biological point of view, was obtained from the log-logistic model fit; and (4) if the four-parameter log-logistic model function failed to fit the concentration-response data. EC50

computation was conducted with the open-source statistical software environment R, version 2.14.2 (http://www.R-project.org).

**Association between *LAMC1* expression levels, EC50 values and patient survival**. Correlations between gene expression and EC50 analyses were calculated in R (Hmisc and ggplot2 libraries, Pearson correlation and corresponding *P* values). To assess correlations and their reproducibility in independent data sets, these analyses were applied to our EC50 values and gene expression data sets from the NCI-60 (53 cell lines, https://discover.nci.nih.gov/cellminer) and the CCLE (The Cancer Cell Line Encyclopedia, using the 38 NCI-60 cell lines that were found in the database)[68]. Data sets were re-formatted and harmonised with in-house code and manual verification. Associations between (mRNAseq) gene expression and overall patient survival in 21 cancers were investigated using data from The Cancer Genome Atlas (TCGA; data available in January 2016). Cox tests were implemented using OncoLnc[69], which used multivariate Cox regressions (R survival library's coxph function, and accounting for sex, age and grade or histology when available). Statistical significance was reported at the level of (nominal) $p = 0.05$.

The Gravendeel, Rembrandt and TCGA data sets for both *LAMC1* gene expression values and Kaplan–Meier patient survival estimation were analysed by means of the GlioVis data portal[70] (http://gliovis.bioinfo.cnio.es/).

**Other statistical analysis**. Graphpad Prism software or Microsoft Excel 2016 were used to perform paired two-tailed Student's *t* test, unless otherwise mentioned. Results are shown as mean values of triplicates ± standard deviation of a representative experiment or as an average of two or more replicated independent experiments.

**Reporting summary**. Further information on research design is available in the Nature Research Reporting Summary linked to this article.

## Data availability

All relevant data supporting the findings of this work are available within the paper and its Supplementary Information files. All other data are available from the corresponding author on request. Fig. 6c and d were generated employing the data sets publicly available at CellMiner™ (https://discover.nci.nih.gov/cellminer) and at The Cancer Cell Line Encyclopedia portals (https://portals.broadinstitute.org/ccle) Supplementary Fig. 7 was generated employing The Cancer Genome Atlas (TCGA) FireBrowse, http://firebrowse.org. Supplementary Fig. 8 was generated using the gene expression data and clinical information from TCGA (http://cancergenome.nih.gov). Supplementary Figs. 9 and 10 were generated using the Glio-Vis data portal (http://gliovis.bioinfo.cnio.es/). Source data are provided with this paper.

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

## Acknowledgements

We thank Winfried Stocker (EuroImmun, Lübeck, Germany) for providing the pTriEX-1-LAMC1 plasmid, Richard Harbottle (DKFZ, Heidelberg, Germany) for the pCAG-S/MAR cloning vector. We are also thankful to the team of the DKFZ Virus Production and Development Unit, in particular Marcus Müller, Barbara Liebetrau and Barbara Leuchs, for helping with virus production and titration and for providing the anti-H-1PV conformational antibody. We are grateful to Jean Rommelaere, Assia Angelova, Marcelo Ehrlich and Steeve Boulant for fruitful discussions. We also thank Sandra Caldeira and Caroline Hadley for critical reading of the manuscript. This study was supported initially by a seeding grant from Institut National du Cancer (INCA) to A.M. and L.B. and at later stages by a grant from ORYX GmbH to A.M. We would also like to express our deepest gratitude to André Welter for his generous donation.

## Author contributions

A.K., T.F., C.B., A.Gr., S.B., T.M. and V.P. designed and performed experiments and analysed the data of these experiments. N.E.A., A.W. and L.B. designed and performed siRNA library screening and assisted in siRNA library screening data analysis; A.K., L.B. and F.A. performed bioinformatic analyses; J.A.H., L.A.R.Y. and H.M. provided and performed the GBM tissue microarray; A.Go. and S.N. provides neurospheres, and assisted with the interpretation of the results obtained from the GlioVis portal; R.R. and B.N. performed the nCounter gene expression analysis; A.K. and A.M. wrote the manuscript and prepared figures with other authors' contribution. A.M. designed, secured funding, participated in data analysis, coordinated and supervised the research. All authors have read and agreed to the published version of the manuscript.

## Funding

## Competing interests

An international patent application protecting some of the results described in this article was submitted in November 2018 with A.K., A.Gr., T.M., T.F. and A.M. as co-inventors. A.M. and N.E.A. are inventors on several H-1PV-related patents/patent applications. The funders had no role in the design of the study, in the collection, analyses, or interpretation of data, in the writing of the manuscript, or in the decision to publish the results. The authors declare no competing interests.
