## [Peer Review File · Nature Communications]

Reviewers' Comments:

Reviewer #1:

Remarks to the Author:

In this interesting and significant manuscript the authors employ an siRNA druggable gene library to provided convincing evidence that laminins, resident in the extracellular matrix (ECM), are intimately involved in the attachment and subsequent entry of the oncolytic rodent protoparvovirus H1-PV into a variety of human tumor cell lines. The authors present a comprehensive set of experiments to support the importance of laminin family members, in particular laminin γ 1, encoded by the LAMC gene, for H1-PV entry, and an extensive set of analyses using panels of tumor cell lines to show that LAMC expression correlates well with permissive to H1-PV infection.

Analysis of cancer patient data shows a negative correlation between laminin γ 1 expression and long-term survival, suggesting that H1-PV oncolytic virotherapy may be significantly efficacious in treating some cancers with a poor prognosis. The siRNA screen also identified AP2M1, a component the complex that catalyzes clathrin-mediated endocytosis, as being required for H1-PV entry, and use a range of inhibitors to support the conclusion that this is the pathway the virus uses to enter HeLa cells.

While overall the manuscript represents a significant contribution to the literature, there are some omissions and sections that need clarification.

Firstly, it is important to note that, as the authors state, the laminins are not integral membrane proteins per se, but reside in the ECM. Thus it is not accurate to describe them as binding H1-PV to the cell membrane, rather they bind the virus at the cell membrane. This distinction is not merely semantic, since the parvoviral paradigm suggests that there should be a transmembrane protein that acts as the viral receptor, and catalyzes the internalization of the virion into a clathrin-coated pit - as indicated by the gap marked as step 2 in the model presented in Figure 6.

Specific Comments

Page 5, line 5: since it is well known that rodent parvovirus infection is absolutely dependent upon sialic acid (SA) - as also shown in Figure 3b of this manuscript - one would expect that any disruption of SA metabolism would score as a hit in a screen of this sort. Is the absence of such hits due to the library not containing siRNAs for any of the SA metabolism enzymes?

Page 5, line 30: what exactly is meant by the term "EGFP signal", used here and in many of the figures? Line 32 refers to "determining the percentage of EGFP-positive cells", whereas line 40 refers to measurement of "EGFP signal intensity per analysed DAPI-stained cells". The authors should define which of these very different measurements is being reported, especially in the Y-axis of several of the panels in Figures 1, 2, 3 and 5.

Page 7, line 22: since they were single-cell cloned, were the CRISPR-derived LAMC1 cells used in the study confirmed for knock out of the gene? If not, why not? If so, they should be described as knock-out rather than knock-down, since their phenotype will be expected to significantly differ from those elicited by partial knock-down of LAMC1 by siRNAs.

Page 10, line 8: was the residual heparin washed out of the culture before virus was added? If not, is it also possible that the heparin is binding to the virus and preventing it from binding sialylated moieties at the cell surface?

Page 15, lines 3-23: Figure 1c neatly summarizes the results of the siRNA screen. However, the supporting information presented in Supplementary Figures 1a, b & c are derivative of the primary data, and do not allow one to drill down on the actual genes involved. It would be very useful to present a ranked list of the 151 genes whose knock-down result in >70% reduction in H1-PV

transduction. These could be colored as in Figure 1a to provide a more comprehensive overview of the results of the screen, and strengthen the rationale for choosing LAMC1 to study further.

Page 16, line 1: since laminin is not an integral membrane protein, it is not likely to be responsible for "cell membrane recognition", rather it would be responsible for virus attachment at the cell membrane (see comment above).

Page 16, line 19: should this be "LAMC1 KO" rather than "KD"? - see comment above.

Page 16, line 38: what is the estimated resolution of the virus:laminin association as measured by this technique?

Page 17, line 44 to page 19, line 1: since the soluble laminin competition studies shown in Supplementary Figures 2 & 3 indicate that many laminin chains other than those containing $\gamma 1$ suppress H1-PV transduction, one must conclude that many different laminin complexes could act as attachment factors in the absence of the LAMC1 gene product. This would likely be the explanation of the substantial residual sensitivity of cells in which siRNA treatment had extensively reduced expression of the targeted laminin, as seen in Figures 2b & c. Thus, in order to claim that laminins "have an essential role in H1-PV infectivity", one would have to knock out all of the laminins expressed in the cell with activity as H1-PV binding proteins, and show that this had a suppressive effect on H1-PV transduction more equivalent to that of neuraminidase than that of a single siRNA targeting LAMC1. To this end, the authors might summarize the expression levels of all the laminin chain mRNAs for which they have this information - for instance, from an RNA-Seq analysis of HeLa cells either conducted by them, or from the literature.

Page 19, lines 30-31: what was the extent of knock down achieved by the single siRNA pool used in Figure 5a? Were any controls run for off-target effects?

Page 19, lines 32-42: while the inhibitors were tested for induction of cytotoxicity, since protoparvoviruses are dependent upon entry into S-phase for the initiation of viral gene expression, controls to discount their disruption of the cell cycle should be run, in order to ensure that the inhibitors are acting as predicted.

Page 21, line 10-20: the possibility that laminins "orchestrate interactions with one or more receptors on the cell surface" is an interesting idea. The authors might wish to discuss this in the light of the paper published by Garcin et al., *Virology* 481: 63-72 (2015), which describes the role of Galectin-3, a potential laminin interactor, in the entry process of Minute Virus of Mice, a rodent protoparvovirus closely related to H1-PV.

Page 25, line 29: what are the units of distance on the X-axis of Figure 3a?

Peter Tattersall February 2020

Reviewer #2:

Remarks to the Author:

Major comments:

1. Although the authors identified that H-1PV infectivity is dependent on LAMC1 expression levels in various cancer cells, they did not show the LAMC1 expression levels in primary cells/tissues as well as H-1PV infectivity/lytic effect in primary cells/tissues. Since TCGA data set (Supplemental Fig. 7) shows some normal tissues express LAMC1 similar to tumors, the authors should evaluate the H-1PV infectivity/lytic effects in human primary cells.
2. Although the authors confirmed that LAMC1 is a key for H-1PV entry, and clathrin-mediated

endocytosis is a key for internalization. Based on RNAi library, there were other genes corresponding to H-1PV lytic effects. In other words, H-1PV anti-tumor effects in vivo may not be dependent on LAMC1 expression levels because for instance, Fig. 4e showed contradict results (NCH125 expressing lower LAMC1 but more H-1PV-dependent lysis than NCH37). The authors should evaluate whether H-1PV-dependent anti-tumor effects in vivo still correspond to LAMC1 expression and need to discuss that.

3. Since murine LAMC1 has high homology to human LAMC1 (more than 90% homology at a.a. level), if rat H-1PV can infect to murine cancer cells/normal tissue through murine LAMC1, the authors can evaluate them in syngeneic mouse models.

Reviewer #3:

Remarks to the Author:

A study that examines, using an siRNA screen of potentially druggable targets in HeLa cells, for genes that alter the efficiency of transduction of cells by an H1 parvovirus expressing a GFP gene in place of the capsid protein gene. They identify 151 genes that reduce transduction when cells are siRNA treated, and 89 that increase transduction. Out of those they pick two that appear to reduce transduction in their assays to follow up on - one encoding the laminin lambda 1 and one encoding the gene for AP2. A lot of the focus of the study (pages of text) is on oncolytic treatment using the virus - which is the particular interest of this group. However, these processes have evolved in the context of natural infection, so a more balanced presentation of the process of natural infection versus oncolysis may be helpful.

1) The reasons for choosing those genes out of all that alter the efficiency of transduction is a little unclear. They then follow up with a number of studies to show that removing laminin reduces cell transduction by 40-60%, suggesting a role in the infectious process or some interference with the processes of replication or expression of the viral genome in the cells; exactly what that might be is unclear. Some other inhibitors are added such as heparin which reduce transduction as well, but the mechanism(s) involved are also unclear. There does not appear to be any evidence for a direct binding of the laminin by the virus, either directly or through sialic acid, in this report.

2) Does the assay being used for the screen (GFP-expression) involve viral replication, and is it dependent on - or enhanced by - the cellular mitosis? How does this correlate with the reduced transduction seen?

3) A confusing issue is the role of the possible receptor binding activities for cell entry and transduction. The virus binds to sialic acids, and that binding is essential for infection of cells (as is shown here as well by NA treatment), but the connection between sialic acid binding and laminin is not clear in the study reported here - removing the sialic acid by NA treatment completely blocks binding and transduction, so there is apparently no laminin binding that does not involve sialic acids? Is the sialic acid on the laminin, and if so is the hypothesis that the glycosylated laminin acts as a specific receptor via sialic acid binding - and that the other cell surface sialic acid does not contribute - or that it makes up the remaining 60% of the susceptibility when laminin is knocked down?

4) The other inhibitors such as heparin are problematic, as those are highly charged and may be binding the virus directly, or causing other indirect effects.

5) The finding that AP2 is required for efficient cell infection is not surprising as clathrin-mediated endocytosis has been shown to be required for infection or transduction of many different parvoviruses, and blocking clathrin-mediated endocytosis by various methods reduces transduction, as expected. While they indicate that this has no effects on the cell health, it is known that blocking clathrin-mediated endocytosis is toxic for the cells, so it is not clear that the reduction in transduction or infection is not due to a reduced mitotic rate of the cells.

6) There are other studies that clearly demonstrate a role for clathrin mediated endocytosis - and AP-2 specifically - in the infection of cells by canine parvovirus bound to the transferrin receptor that are not cited.

7) The brief analysis of Rab5 and Rab7 co-localization - those studies are not easy to conduct as the Rab overexpression alters trafficking within endosomes and membranes within the cells. It is possible to show the specific co-localization, but that needs to be conducted with careful quantification and dynamic tracking, so the results shown here are not definitive proof of the intracellular trafficking pathway - although that is a likely process.

8) The connection of laminin level to cancer susceptibility seems possible, and is likely worth following up on in future studies.

9) This manuscript was quite hard to read - being very dense overall and quite hard to follow in places. A careful re-write to focus on the mechanisms being investigated here would help. Using strategic subheadings and reorganization would be helpful in many places.

Deutsches Krebsforschungszentrum | M123 | PF 101949 | 69009 Heidelberg

Reviewer #1 (Remarks to the Author):

In this interesting and significant manuscript the authors employ an siRNA druggable gene library to provided convincing evidence that laminins, resident in the extracellular matrix (ECM), are intimately involved in the attachment and subsequent entry of the oncolytic rodent protoparvovirus H1-PV into a variety of human tumor cell lines. The authors present a comprehensive set of experiments to support the importance of laminin family members, in particular laminin γ 1, encoded by the LAMC gene, for H1-PV entry, and an extensive set of analyses using panels of tumor cell lines to show that LAMC expression correlates well with permissive to H1-PV infection.

Analysis of cancer patient data shows a negative correlation between laminin γ 1 expression and long-term survival, suggesting that H1-PV oncolytic virotherapy may be significantly efficacious in treating some cancers with a poor prognosis. The siRNA screen also identified AP2M1, a component the complex that catalyzes clathrin-mediated endocytosis, as being required for H1-PV entry, and use a range of inhibitors to support the conclusion that this is the pathway the virus uses to enter HeLa cells. While overall the manuscript represents a significant contribution to the literature, there are some omissions and sections that need clarification.

**Laboratory of Oncolytic Virus
Immuno-Therapeutics (LOVIT)**
F011
Priv. Doz. Dr. Antonio Marchini
Principal Investigator

Im Neuenheimer Feld 242
69120 Heidelberg
Telefon +49 6221 42-4969
Telefax +49 6221 42-4962
a.marchini@dkfz.de
www.dkfz.de

Heidelberg, 10.11.2020

Authors: We thank the reviewer for his positive feedback and addressed his remaining criticism (see below).

Firstly, it is important to note that, as the authors state, the laminins are not integral membrane proteins per se, but reside in the ECM. Thus it is not accurate to describe them as binding H1-PV **to** the cell membrane, rather they bind the virus **at** the cell membrane. This distinction is not merely semantic, since the parvoviral paradigm suggests that there should be a transmembrane protein that acts as the viral receptor, and catalyzes the internalization of the virion into a clathrin-coated pit - as indicated by the gap marked as step 2 in the model presented in Figure 6.

Authors: We agree with the reviewer that a transmembrane protein may act as (co)-receptor and mediate H-1PV cell entry. We have corrected all instances in the manuscript and indicate that laminins are involved in H-1PV binding at the cell surface and play a fundamental role in virus cell entry.

Specific Comments

Page 5, line 5: since it is well known that rodent parvovirus infection is absolutely dependent upon sialic acid (SA) - as also shown in Figure 3b of this manuscript - one would expect that any disruption of SA metabolism would score as a hit in a screen of this sort. Is the absence of such hits due to the library not containing siRNAs for any of the SA metabolism enzymes?

Stiftung des öffentlichen Rechts

Stiftungsvorstand
Prof. Dr. med. Michael Baumann
Prof. Dr. rer. pol. Josef Puchta

Deutsche Bank Heidelberg
IBAN: DE09 6727 0003 0015 7008 00
BIC (SWIFT): DEUT DES M672

Deutsche Bundesbank Karlsruhe
IBAN: DE39 6600 0000 0067 0019 02
BIC (SWIFT): MARK DEF 1660

Authors: We thank the reviewer for these comments. We searched whether the library used for the siRNA library screening contained enzymes involved in SA metabolism. We found that the library contained glucosamine (UDP-N-acetyl)-2-epimerase/N-acetylmannosamine kinase (*GNE*) as the only gene involved in SA biosynthesis. In agreement with the idea that SA plays a central role in H-1PV cell attachment and entry, siRNA mediated knock-down of *GNE* dramatically reduced H-1PV transduction. This new information has been added at page 16 (lines 2-8).

Page 5, line 30: what exactly is meant by the term "EGFP signal", used here and in many of the figures? Line 32 refers to "determining the percentage of EGFP-positive cells", whereas line 40 refers to measurement of "EGFP signal intensity per analysed DAPI-stained cells". The authors should define which of these very different measurements is being reported, especially in the Y-axis of several of the panels in Figures 1, 2, 3 and 5.

Authors: We are sorry for the confusion created. For Fig. 1 both parameters have been analysed. First, the EGFP positive cells were detected and, in these cells, the EGFP signal intensity measured. This information has been added in the M&M section at page 6 (lines 8-9). For the other figures the number of EGFP positive cells were calculated. We corrected the Y axis in the panels concerned of Fig. 2, 3 and 5 (now 6), accordingly.

Page 7, line 22: since they were single-cell cloned, were the CRISPR-derived LAMC1 cells used in the study confirmed for knock out of the gene? If not, why not? If so, they should be described as knock-out rather than knock-down, since their phenotype will be expected to significantly differ from those elicited by partial knock-down of LAMC1 by siRNAs.

Authors: We could not achieve a complete knock-out of the *LAMC1* gene in HeLa cells by CRISP-Cas9 technology. We give this information at page 17 lines 18-20. A possible explanation for this is that HeLa cells are polyploidy in their genome and may contain multiple copies of the *LAMC1* gene which is challenging to target in full. Another possibility is that the *LAMC1* gene is essential for the viability of the cells and therefore we select only those clones that maintain a residual *LAMC1* activity. We added a higher exposure of the WB showing that in HeLa LAMC1 KD cells laminin γ 1 chain levels are reduced but not completely absent (Figure 2 f). Reduced laminin γ 1 correlated with a reduction of H-1PV binding/entry activity at the cell surface that was rescued by transient transfection of the *LAMC1* gene in these cells.

Page 10, line 8: was the residual heparin washed out of the culture before virus was added? If not, is it also possible that the heparin is binding to the virus and preventing it from binding sialylated moieties at the cell surface?

Authors: Heparin was not washed away before H-1PV infection so we found this criticism valid. Therefore, in the revised version of our manuscript, we performed control experiments in which we clearly showed that heparin cannot bind directly to H-1PV capsid. These results have been described at page 19 lines 21-24 and Fig. 5c (with relative figure legend and M&M).

Page 15, lines 3-23: Figure 1c neatly summarizes the results of the siRNA screen. However, the supporting information presented in Supplementary Figures 1a, b & c are derivative of the primary data, and do not allow one to drill down on the actual genes involved. It would be very useful to present a ranked list of the 151 genes whose knock-down result in >70% reduction in H1-PV transduction. These could be colored as in Figure 1a to provide a more comprehensive overview of the results of the screen, and strengthen the rationale for choosing LAMC1 to study further.

Authors: Our goal was to identify factors required in cell attachment and entry. To this end, in the present study we focused on two protein classes: extracellular matrix protein and transmembrane signal receptor. A list of the genes present in these two categories has been provided (new Suppl. Tab. 1). The rationale for the selection of *LAMC1* has been explained in details at page 16 lines 23-40.

Page 16, line 1: since laminin is not an integral membrane protein, it is not be likely to be responsible for "cell membrane recognition", rather it would be responsible for virus attachment at the cell membrane (see comment above).

Authors: thank you for your comment. This has been corrected in "cell attachment and entry".

Page 16, line 19: should this be "LAMC1 KO" rather than "KD"? - see comment above.

Authors: please see our answer above.

Page 16, line 38: what is the estimated resolution of the virus:laminin association as measured by this technique?

Authors: The imaging was performed at the resolution limit of 200 nm, information added in the M&M section at page 8 (lines 33-35).

Page 17, line 44 to page 19, line 1: since the soluble laminin competition

studies shown in Supplementary Figures 2 & 3 indicate that many laminin chains other than those containing $\gamma 1$ suppress H1-PV transduction, one must conclude that many different laminin complexes could act as attachment factors in the absence of the LAMC1 gene product. This would likely be the explanation of the substantial residual sensitivity of cells in which siRNA treatment had extensively reduced expression of the targeted laminin, as seen in Figures 2b & c. Thus, in order to claim that laminins "have an essential role in H1-PV infectivity", one would have to knock out all of the laminins expressed in the cell with activity as H1-PV binding proteins, and show that this had a suppressive effect on H1-PV transduction more equivalent to that of neuraminidase than that of a single siRNA targeting LAMC1. To this end, the authors might summarize the expression levels of all the laminin chain mRNAs for which they have this information - for instance, from an RNA-Seq analysis of HeLa cells either conducted by them, or from the literature.

Authors: Our results indicate that H-1PV can bind different laminins containing the laminin $\gamma 1$ chain. These results suggest that multiple laminins may be involved in H-1PV cell attachment and entry. The silencing of the *LAMC1* (or *LAMB1* gene) did strongly reduce H-1PV infection but it did not block it at the same level than NA treatment. Silencing of multiple laminin chain encoding genes may be more efficient in blocking H-1PV infection. However, when we tried to silence multiple laminin chain encoding genes, we found that this was deleterious for the viability of the cells.

Residual H-1PV binding/transduction activity can also result from an incomplete silencing of the gene leading to residual protein levels of laminin $\gamma 1$ chain still expressed in the cells (as our Western Blot analyses indicate by exposing the Blots for longer times). Furthermore, we cannot completely rule out that in addition to laminins the virus may also use alternative factors for its cell attachment and entry. These results have been discussed more extensively at page 22 (lines 21-28).

Following the suggestion of the reviewer the expression levels of laminin chain mRNAs in HeLa cells have been summarized in Suppl. Fig. 3a and in the results section (page 18, lines 8-11).

Page 19, lines 30-31: what was the extent of knock down achieved by the single siRNA pool used in Figure 5a? Were any controls run for off-target effects?

Page 19, lines 32-42: while the inhibitors were tested for induction of cytotoxicity, since protoparvoviruses are dependent upon entry into S-phase for the initiation of viral gene expression, controls to discount their disruption of the cell cycle should be run, in order to ensure that the inhibitors are acting as predicted.

Authors: These results are no longer part of the manuscript.

Page 21, line 10-20: the possibility that laminins "orchestrate interactions with one or more receptors on the cell surface" is an interesting idea. The authors might wish to discuss this in the light of the paper published by Garcin et al., Virology 481: 63-72 (2015), which describes the role of Galectin-3, a potential laminin interactor, in the entry process of Minute Virus of Mice, a rodent protoparvovirus closely related to H1-PV.

Authors: Thank you very much for the suggestion. This has been discussed at page 23 (lines 1-10) and suggested reference added.

Page 25, line 29: what are the units of distance on the X-axis of Figure 3a?

Authors. The units are the RGB values. This has been specified in the figure.

**Reviewer #2 (Remarks to the Author):**

Major comments:

1. Although the authors identified that H-1PV infectivity is dependent on LAMC1 expression levels in various cancer cells, they did not show the LAMC1 expression levels in primary cells/tissues as well as H-1PV infectivity/lytic effect in primary cells/tissues. Since TCGA data set (Supplemental Fig. 7) shows some normal tissues express LAMC1 similar to tumors, the authors should evaluate the H-1PV infectivity/lytic effects in human primary cells.

Authors: We agree with this comment. It is known that normal tissue also expresses laminins. H-1PV can also infect normal non-transformed cells although most of the times these infections do not lead to production of progeny virus particles, as normal cells probably lack important determinants needed for H-1PV replication (reviewed in Angelova et al. 2015, new added reference number 68).

In the revised version of the manuscript, we performed a tissue microarray including 110 different patient biopsies obtained from 61 primary and 49 recurrent GBMs and analyzed the levels of laminin γ 1 chain. Interestingly, we confirmed at the protein level that different tumours can express different levels of LAMC1, and we speculate that tumours with higher levels of LAMC1 may be more susceptible to H-1PV infection than tumours with low levels of LAMC1 expression. However, the virus needs other factors acting at the entry and post-entry levels for a productive infection. In our opinion, only the analysis of a number of modulators (positive or negative) on tumour specimens can identify those patients most likely responding to H-1PV treatment (discussed at page 24 line 44 and at page 25 lines 1-15).

2. Although the authors confirmed that LAMC1 is a key for H-1PV entry, and clathrin-mediated endocytosis is a key for internalization. Based on RNAi library, there were other genes corresponding to H-1PV lytic effects. In other words, H-1PV anti-tumor effects in vivo may not be dependent on LAMC1 expression levels because for instance, Fig. 4e showed contradict results (NCH125 expressing lower LAMC1 but more H-1PV-dependent lysis than NCH37). The authors should evaluate whether H-1PV-dependent anti-tumor effects in vivo still correspond to LAMC1 expression and need to discuss that.

Authors: Thank you for your comment. We agree that other genes are most likely involved and that the single analysis of laminins is likely insufficient to predict the outcome of virus treatment (see our previous answer to your comment). Further studies are required to prove whether other putative genes identified in our siRNA library screen are equally important for H-1PV life cycle. This has been discussed in the new version of the manuscript at page 24 (line 44) and page 25 (lines 1-15).

3. Since murine LAMC1 has high homology to human LAMC1 (more than

90% homology at a.a. level), if rat H-1PV can infect to murine cancer cells/normal tissue through murine LAMC1, the authors can evaluate them in syngeneic mouse models.

Authors: It is known that mouse cancer cells are not permissive to H-1PV infection. The reasons for this resistance are not clear but the block is most of the time at post-entry level. Experiments with immunocompetent rat models may be interesting to further validate our results obtained in cell culture models in animals. However, in our opinion this is a new project in itself which goes beyond the initial objective of the present study that was to identify H-1PV cell attachment factors. It would require the establishment of rat cancer cell lines overexpressing LAMC1 or in which LAMC1 gene has been knocked-down (or out). Furthermore, we would need to apply for and obtain authorization to perform these experiments in animals; a long and rightly so, highly regulated procedure in Germany. At this stage, we believe that the discovery that laminins are important modulators of H-1PV cell attachment and entry is important enough to justify publication without additional delays and sacrificing animals.

**Reviewer #3 (Remarks to the Author):**

A study that examines, using an siRNA screen of potentially druggable targets in HeLa cells, for genes that alter the efficiency of transduction of cells by an H1 parvovirus expressing a GFP gene in place of the capsid protein gene. They identify 151 genes that reduce transduction when cells are siRNA treated, and 89 that increase transduction. Out of those they pick two that appear to reduce transduction in their assays to follow up on - one encoding the laminin lambda 1 and one encoding the gene for AP2. A lot of the focus of the study (pages of text) is on oncolytic treatment using the virus - which is the particular interest of this group. However, these processes have evolved in the context of natural infection, so a more balanced presentation of the process of natural infection versus oncolysis may be helpful.

1) The reasons for choosing those genes out of all that alter the efficiency of transduction is a little unclear. They then follow up with a number of studies to show that removing laminin reduces cell transduction by 40-60%, suggesting a role in the infectious process or some interference with the processes of replication or expression of the viral genome in the cells; exactly what that might be is unclear. Some other inhibitors are added such as heparin which reduce transduction as well, but the mechanism(s) involved are also unclear. There does not appear to be any evidence for a direct binding of the laminin by the virus, either directly or through sialic acid, in this report.

Authors: We took this comment very seriously. In the revised version of the manuscript, we are very pleased to provide the evidence requested by this reviewer. With a completely new set of experiments we successfully show that H-1PV directly binds to laminins. These results are described in the new paragraph "H-1PV directly binds to laminins through sialic acid within the heparin binding site" at page 19 and also illustrated in the new Fig. 5 (with relative figure legend and M&M). We were able to recapitulate in vitro by ELISA, the results obtained in cell culture models and to shed light on the mechanisms underlying laminins/H-1PV interactions. We showed that H-1PV binds directly laminins without the involvement of other cellular factors. In agreement with previous results, we show that the virus preferentially binds to laminins containing the laminin γ 1 chain. We also showed that neuraminidase treatment abolishes these interactions, indicating that the sialic acid present within laminins is responsible for the interaction. Furthermore, we also confirmed that heparin, by binding to laminins, decreases H-1PV/laminins interactions without binding itself to H-1PV capsid. We hope that these efforts are appreciated by the reviewer.

2) Does the assay being used for the screen (GFP-expression) involve viral replication, and is it dependent on - or enhanced by - the cellular mitosis? How does this correlate with the reduced transduction seen?

Authors: The recombinant H-1PV harbouring the EGFP reporter gene used for the screen is a propagation deficient virus (cannot replicate) because the EGFP gene replaces part of the VP gene unit encoding for virus capsid proteins (information added at page 5 lines 37-38). However, the EGFP gene is under the control of the natural H-1PV P38 promoter which is activated by the NS1 viral protein. Therefore, EGFP signal is a direct measurement of the ability of the virus to transduce these cells. As activation of P4 promoter which regulates the expression of NS1 protein requires factors expressed in S phase of the cell cycle (e.g. reference N.17), the recombinant virus can only transduce those cells that are actively proliferating (in a similar manner as wild type H-1PV). The siRNAs used to target the laminins encoding genes did not alter the viability of the cells.

3) A confusing issue is the role of the possible receptor binding activities for cell entry and transduction. The virus binds to sialic acids, and that binding is essential for infection of cells (as is shown here as well by NA treatment), but the connection between sialic acid binding and laminin is not clear in the study reported here - removing the sialic acid by NA treatment completely blocks binding and transduction, so there is apparently no laminin binding that does not involve sialic acids? Is the sialic acid on the laminin, and if so is the hypothesis that the glycosylated laminin acts as a specific receptor via sialic acid binding - and that the other cell surface sialic acid does not contribute - or that it makes up the remaining 60% of the susceptibility when laminin is knocked down?

Authors: This has been clarified in the revised version of the manuscript (see our previous answer to comment 1).

4) The other inhibitors such as heparin are problematic, as those are highly charged and may be binding the virus directly, or causing other indirect effects.

Authors: This has been clarified in the revised version of the manuscript (see our previous answer to comment 1).

5) The finding that AP2 is required for efficient cell infection is not surprising as clathrin-mediated endocytosis has been shown to be required for infection or transduction of many different parvoviruses, and blocking clathrin-mediated endocytosis by various methods reduces transduction, as expected. While they indicate that this has no effects on the cell health, it is known that blocking clathrin-mediated endocytosis is toxic for the cells, so it is not clear that the reduction in transduction or

infection is not due to a reduced mitotic rate of the cells.

6) There are other studies that clearly demonstrate a role for clathrin mediated endocytosis - and AP-2 specifically - in the infection of cells by canine parvovirus bound to the transferrin receptor that are not cited.

7) The brief analysis of Rab5 and Rab7 co-localization - those studies are not easy to conduct as the Rab overexpression alters trafficking within endosomes and membranes within the cells. It is possible to show the specific co-localization, but that needs to be conducted with careful quantification and dynamic tracking, so the results shown here are not definitive proof of the intracellular trafficking pathway - although that is a likely process.

Authors: We thank the reviewer for these comments. The paragraph describing that H-1PV uses clathrin mediated endocytosis for its entry has been removed from this manuscript. These results have been recently published elsewhere (new ref. N. 31).

8) The connection of laminin level to cancer susceptibility seems possible, and is likely worth following up on in future studies.

Authors: Thank you for the positive comment. In this manuscript we added a new paragraph entitled "Laminin γ 1 chain is differentially expressed in primary and recurrent GBM biopsies" supporting the idea that it is worth to continue with this line of research in the future.

9) This manuscript was quite hard to read - being very dense overall and quite hard to follow in places. A careful re-write to focus on the mechanisms being investigated here would help. Using strategic subheadings and reorganization would be helpful in many places.

Authors: We followed this suggestion and provide new paragraphs focusing on the mechanisms. We are confident that with these changes we have improved our original manuscript.

Reviewers' Comments:

Reviewer #1:

Remarks to the Author:

This is a significantly more focused version of the original submission, in which the authors have addressed all of the comments and concerns of the reviewers.

They have removed the section on entry via clathrin-mediated endocytosis, which has been extended and published elsewhere. The revised manuscript concentrates on the role of the laminins in H-1PV cell surface attachment, and now contains an expanded the sections on the role of laminin γ 1, and the potential use of tumor laminin expression profiling in stratifying patients for treatment with this particular oncolytic virus.

They have added an ELISA for measuring direct binding of H-1PV to various laminins and used this assay to demonstrate competition for such binding by heparin.

They also report that the only sialic acid metabolism gene for which there is an siRNA pool in their library is GNE, and that GNE knock-down has a similar negative effect on H-1PV transduction/infection as neuraminidase treatment, supporting the essential role of sialic acid moieties in the interaction of H-1PV with laminin γ 1 revealed the ELISA experiments.

The text inserted in the revised manuscript contained some passages that the authors might consider clarifying -

Page 22, lines 17-18: "was found to privilege interactions with" - do you mean "was found to interact preferentially with"?

Page 23, line 12: "prerogative" - do you mean "property"?

Reviewer #2:

Remarks to the Author:

The authors propose H-1PV as an oncolytic virus agent and identified LAMC1 dependent infection mechanism. However, there was no significant different of LAMC1 expression between normal tissue and GBM tumors in additional TMA data. They referred an article mentioning no/limited replication in normal/primary cells, but there is no evidence by authors in revised manuscript. The authors should evaluate the H-1PV infectivity/lytic effects in human primary cells (e.g. fibroblast, endothelial cell, epithelial cell, adipocytes: all available at ATCC).

Following above comment, the authors did not address how this infection mechanism correlates with outcome (anti-tumor activity). Without in vivo studies, these findings have no impact in the oncolytic virotherapy field.

Reviewer #3:

Remarks to the Author:

A revised manuscript on the study that shows a role for Laminin 1 in the infection or transduction of cells with the H1 parvovirus. The revision is well presented and addresses virtually all of the comments of the previous review.

While generally well conducted and I have few issues with the studies and the data, there are some key remaining issues with the presentation, which can be addressed by strategic rewriting.

1) The claim (or implication) that the virus binds to the Laminin 1. It is clear from the data that the virus binds to the sialic acid on the Laminin, so the sialic acid is still the main receptor, as has been

known for a number of years. This study therefore reveals that one protein is an important sialic acid-display vehicle, allowing it to be transported into the cell in some as-yet defined process. This is therefore distinct from the model described in the introduction where the sialic acid acts as a non-specific ligand that allow the virus to associate with the cell surface and then to transfer to another receptor for cell infection. The title therefore might be "infects cells by binding sialic acid on Laminin-1 for cell attachment and entry". Since the Laminin 1 knock out leaves some remaining binding and entry occurring, it appears that other glycoproteins (or other sialic acid bearing molecules) exist.

This type of receptor interactions resembles those of other viruses bind sialic acid, where certain host glycoproteins are identified as the primary glycoprotein mediating entry - again those seem to be hard to pin down to the specific processes involved.

2) The role and effect of heparin treatment. The apparent effect of heparin on the infection is used to define the possible site of the sialic acid on the Laminin 1. This seems like an over-interpretation; heparin is a highly charged molecule and may bind non-specifically to a number of molecules through a purely ionic interaction. Without direct proof of the role of the specific glycan this should be toned down, and the likely site of binding (presumably site of glycosylation) not claimed with any degree of certainty.

RE: Our manuscript: NCOMMS-20-02837

Previous title: "Oncolytic H-1 parvovirus hijacks laminins for cell attachment and entry"

New Title: "Oncolytic H-1 parvovirus binds to sialic acid on laminins for cell attachment and entry" by
Kulkarni et al.

Please find below our point-to point response letter to reviewers' comments

Reviewer #1 (Remarks to the Author):

This is a significantly more focused version of the original submission, in which the authors have addressed all of the comments and concerns of the reviewers.
They have removed the section on entry via clathrin-mediated endocytosis, which has been extended and published elsewhere. The revised manuscript concentrates on the role of the laminins in H-1PV cell surface attachment, and now contains an expanded the sections on the role of laminin $\gamma 1$, and the potential use of tumor laminin expression profiling in stratifying patients for treatment with this particular oncolytic virus.
They have added an ELISA for measuring direct binding of H-1PV to various laminins and used this assay to demonstrate competition for such binding by heparin.
They also report that the only sialic acid metabolism gene for which there is an siRNA pool in their library is GNE, and that GNE knock-down has a similar negative effect on H-1PV transduction/infection as neuraminidase treatment, supporting the essential role of sialic acid moieties in the interaction of H-1PV with laminin $\gamma 1$ revealed the ELISA experiments.

Authors: We thank the reviewer for an insightful and helpful review that improved the quality of our manuscript.

The text inserted in the revised manuscript contained some passages that the authors might consider clarifying -

Page 22, lines 17-18: "was found to privilege interactions with" - do you mean "was found to interact preferentially with"?

Page 23, line 12: "prerogative" - do you mean "property"?

Authors: We agree with these comments and corrected the text accordingly.

Reviewer #2 (Remarks to the Author):

The authors propose H-1PV as an oncolytic virus agent and identified LAMC1 dependent infection mechanism.

We are not proposing in this study a novel oncolytic virus agent; the natural oncotropism of H1-PV, specific oncotoxicity as well as its ability to trigger anticancer immune responses are well established by more than 50 years of research at the preclinical level (reviewed in Angelova et al. 2019). Most importantly, H-1PV has been assessed as a monotherapy in two phase I and II/a clinical trials in patients with glioblastoma (NCT01301430) and pancreatic carcinomas (NCT02653313). The results of these clinical studies show that virus treatment is safe, well tolerated and associated with surrogate evidence of anticancer efficacy including improved overall survival in comparison with historical controls.

These promising preclinical and clinical results prompt us to study the early steps of infection of H-1PV to identify the cellular factors that mediate H-1PV cell attachment and entry.

Cited Ref: Angelova, A. & Rommelaere, J. Immune System Stimulation by Oncolytic Rodent Protoparvoviruses. *Viruses* 11, doi:10.3390/v11050415 (2019).

However, there was no significant different of LAMC1 expression between normal tissue and GBM tumors in additional TMA data.

We fear that this concern from the reviewer stems from a misunderstanding. It is likely that the reviewer understood primary as normal tissues instead than as **primary tumours**.

The results presented in the former version of our manuscript were all derived analyzing biopsies from glioblastoma patients (namely 61 **primary glioblastoma tumours** and 49 **recurrent glioblastoma tumours**). No results **from normal tissues** were presented in the previous version of our manuscript. Our experiment indicates that laminin- γ chain is differentially expressed among tumors. This is of utmost clinical relevance, as clinical studies demonstrate the most effective route of administration of H-1PV (Geletneky et al 2017) is intratumoral. Our results showing the important role of laminins in H-1PV infection, support the idea that the tumours that express high levels of LAMC1 may respond better to H-1PV treatment than those in which LAMC1 is absent or expressed at very low levels.

However, the comment of the reviewer prompted us to assess laminin- γ chain protein levels in normal tissues. In this revised version of the manuscript, we also included the analysis of the laminin- γ chain protein levels **in normal tissues** from brain, liver and tonsil for a total of 12 biopsies. Our results indicate a significant difference between the laminin- γ chain protein levels in normal tissues and tumour samples, with the tumours showing higher levels of the protein than normal tissues.

They referred an article mentioning no/limited replication in normal/primary cells, but there is no evidence by authors in revised manuscript. The authors should evaluate the H-1PV infectivity/lytic effects in human primary cells (e.g. fibroblast, endothelial cell, epithelial cell, adipocytes: all available at ATCC).

We are confused by this remark. A significant number of different studies have addressed the effects of H-1PV infection in normal cells from different tissues and showed its innocuousness, *e. g.* our laboratory showed that H-1PV infection does not kill normal astrocytes, melanocytes, oral fibroblasts, foreskin fibroblasts while is very efficient in killing cancer cells (Li et al EMBO Mol. Med. 2013, Supportive

Information Fig. 2). In the previous version of the manuscript, we refer to **a review** article (Angelova et al. 2015) which summarizes all these preclinical evidence because the list of these articles would be too long.

We have prepared a file (added as additional material) reviewing part of this literature. It is an incomplete selection of studies but we hope it may convince the reviewer that this issue has been exhaustively addressed in the past. We also note that the above mentioned clinical studies show the safety of H-1PV treatment which validates preclinical evidence.

Nevertheless, in our new version of the manuscript, to give an idea to the readers that there is an extensive literature about the safety of H-1PV in normal cells, in addition to the review previously cited in our manuscript, we also included two other references as examples showing the assessment of H-1PV in normal cells (Ref. 54 and 55). Furthermore, we have also analyzed the LAMC1 mRNA levels **in normal astrocytes** by nanostring analysis. These cells express low levels of *LAMC1* and are resistant to H-1PV oncolytic activity. These results are shown in the new Fig. 6e.

Following above comment, the authors did not address how this infection mechanism correlates with outcome (anti-tumor activity). Without in vivo studies, these findings have no impact in the oncolytic virotherapy field.

Once again, we are unsure as to what the Reviewer means here and fear that there is a misunderstanding about the main objective of our study. Our goal was to characterize the early steps of infection of H-1PV, and in particular the mechanisms underlying virus attachment at the cell surface and its cell entry. We do not investigate the mechanisms related to virus replication and oncolysis, which occur after virus entry and certainly require the involvement of other cellular factors, partly characterized so far (reviewed in Nuesch et al.). Infection and oncolytic activity are two distinct phases of the virus life cycle.

Notwithstanding, our results in cancer cells (53 cancer cell lines, primary GBM cultures and organoids) do reveal a direct correlation between LAMC1 expression levels and virus oncolytic activity. This correlation is plausible as virus entry is the first important (and obligatory) step preceding replication and oncolysis. Thus, we feel that with this large set of experiments we have addressed how the infection mechanism correlates with anti-tumour activity.

We do not agree with the Reviewer 2's comment that in vivo studies are necessary. Beyond any ethical and legal reservations (3Rs) regarding animal experimentation in this case, we reinforce here that our goal is to characterize the role of laminins in H-1PV cell attachment and entry and not the mechanisms underlying H-1PV-mediated oncosuppression (which are surely dependent on many other factors including the role of the immune-system). We believe that the manuscript focus should remain on the mechanisms of H-1PV entry

Cited Ref: Nuesch, J. P., Lacroix, J., Marchini, A. & Rommelaere, J. Molecular pathways: rodent parvoviruses-mechanisms of oncolysis and prospects for clinical cancer treatment. Clin Cancer Res 18, 3516-3523, doi:10.1158/1078-0432.CCR-11-2325 (2012).

Reviewer #3 (Remarks to the Author):

A revised manuscript on the study that shows a role for Laminin 1 in the infection or transduction of cells with the H1 parvovirus. The revision is well presented and addresses virtually all of the comments of the previous review.

Authors: Thank you very much for your positive feedback and for the appreciation that we have addressed your previous concerns. We thank you for your excellent comments/suggestions that improved the quality of our manuscript.

While generally well conducted and I have few issues with the studies and the data, there are some key remaining issues with the presentation, which can be addressed by strategic rewriting.

1) The claim (or implication) that the virus binds to the Laminin 1. It is clear from the data that the virus binds to the sialic acid on the Laminin, so the sialic acid is still the main receptor, as has been known for a number of years. This study therefore reveals that one protein is an important sialic acid-display vehicle, allowing it to be transported into the cell in some as-yet defined process. This is therefore distinct from the model described in the introduction where the sialic acid acts as a non-specific ligand that allow the virus to associate with the cell surface and then to transfer to another receptor for cell infection. The title therefore might be "infects cells by binding sialic acid on Laminin-1 for cell attachment and entry". Since the Laminin 1 knock out leaves some remaining binding and entry occurring, it appears that other glycoproteins (or other sialic acid bearing molecules) exist. This type of receptor interactions resembles those of other viruses bind sialic acid, where certain host glycoproteins are identified as the primary glycoprotein mediating entry - again those seem to be hard to pin down to the specific processes involved.

Authors: We thank the reviewer for her/his criticism and interesting point raised. We believe that we have recognized sufficiently the critical role of SA in H-1PV infection in many sections of our manuscript (even confirming previous results and providing new ones for that). However, we are also convinced that this does not detract from the novelty of the manuscript that sheds light, for the first time, on a family of proteins that mediate H-1PV-cell surface interaction via SA moieties. This discovery has potential clinical relevance as these glycoproteins could serve as prediction markers for the selection of those tumours more susceptible to H-1PV infection.

We have revised the text to be in line with the comment of the reviewer. These are the main changes:

Title. As suggested by the reviewer we revised the title emphasizing the role of SA. The new title reads like that: **"Oncolytic H-1 parvovirus binds to sialic acid on laminins for cell attachment and entry"**

Introduction: It was not our intention to suggest any model in the introduction. We revised the aim of our study and have rewritten it in more general terms (page 4 lines 2-8).

Discussion: we have revised the text clarifying the importance of sialylated laminins in H-1PV infection and the possibility that other glycoprotein could also participate in H-1PV cell attachment and entry (page 22 lines 10-16).

2) The role and effect of heparin treatment. The apparent effect of heparin on the infection is used to define the possible site of the sialic acid on the Laminin 1. This seems like an over-interpretation; heparin is a highly charged molecule and may bind non-specifically to a number of molecules through a purely ionic interaction. Without direct proof of the role of the specific glycan this should be toned down, and the likely site of binding (presumably site of glycosylation) not claimed with any degree of certainty.

Authors. Thank you for this valuable comment. We fully agree and we have now addressed this issue in the revised manuscript (page 24 lines 21-28) and toned down the effect of heparin treatment in the results section (page 19, line 31).